# Geometric Algebra Planes:
# Convex Implicit Neural Volumes

**Irmak Sivgin** [* 1]    **Sara Fridovich-Keil** [* 1 2]    **Gordon Wetzstein** [1]    **Mert Pilanci** [1]

## Abstract

Volume parameterizations abound in recent literature, encompassing methods from classic voxel grids to implicit neural representations. While implicit representations offer impressive capacity and improved memory efficiency compared to voxel grids, they traditionally require training through nonconvex optimization, which can be slow and sensitive to initialization and hyperparameters. We introduce GA-Planes, a novel family of implicit neural volume representations inspired by Geometric Algebra that can be trained using convex optimization, addressing the limitations of nonconvex methods. GA-Planes models generalize many existing representations including any combination of features stored in tensor basis elements followed by a neural feature decoder, and can be adapted to convex or nonconvex training as needed for various inverse problems. In the 2D setting, we prove GA-Planes models are equivalent to a low-rank plus low-resolution matrix factorization that outperforms the classic low-rank plus sparse decomposition for fitting a natural image. In 3D, GA-Planes models exhibit competitive expressiveness, model size, and optimizability across tasks such as radiance field reconstruction, 3D segmentation, and video segmentation. Code is available at https://github.com/sivginirmak/Geometric-Algebra-Planes.

## 1. Introduction

Volumes are everywhere—from the world we live in to the videos we watch to the organs and tissues inside our bodies. In recent years tremendous progress has been made in modeling these volumes using measurements and computation (Tewari et al., 2022), to make them accessible for downstream tasks in applications including manufacturing (Intwala & Magikar, 2016; Šlapak et al., 2024), robotic navigation (Ming et al., 2024; Wijayathunga et al., 2023), entertainment and culture (Liu et al., 2024; Croce et al., 2023), and medicine (Udupa & Herman, 1999; Masero et al., 2002; Richter et al., 2024; Xu et al., 2024). All methods that seek to model a volume face a three-way tradeoff between *model size*, which determines hardware memory requirements, *expressiveness*, which determines how faithfully the model can represent the underlying volume, and *optimizability*, which captures how quickly and reliably the model can learn the volume from measurements. Certain applications place stricter requirements on model size (e.g. for deployment on mobile or edge devices), expressiveness (e.g. resolution required for medical diagnosis or safe robotic navigation), or optimizability (e.g. for interactive applications), but all stand to benefit from improvements to this three-way pareto frontier.

Many existing strategies have been successfully applied at different points along this pareto frontier; some representative examples from computer vision are summarized in Appendix A.1. Our goal is to maintain or surpass the existing pareto frontier of model size and expressiveness while improving optimization stability through convexity.

Our approach introduces *convex* and *semiconvex* reformulations of the volume modeling optimization process that apply to a broad class of volume models we call *Geometric Algebra Planes*, or GA-Planes for short. We adopt the term *semiconvex* for Burer-Monteiro (BM) factorizations of a convex objective, as introduced in (Sahiner et al., 2024), within the context of convex neural networks. BM factorized problems have the property that every local minimum is globally optimal (Sahiner et al., 2024).

GA-Planes is a mixture-of-primitives model that generalizes several existing volume models including voxels and tensor factorizations. Most importantly, most models in this family can be formulated for optimization by a convex program, as long as the objective function (to fit measurements of the volume) is convex. At the same time, *any* GA-Planes model

---

*Equal contribution  [1]Department of Electrical Engineering, Stanford University, CA, USA  [2]School of Electrical & Computer Engineering, Georgia Institute of Technology, GA, USA. Correspondence to: Mert Pilanci <pilanci@stanford.edu>.

*Proceedings of the 42$^{nd}$ International Conference on Machine Learning*, Vancouver, Canada. PMLR 267, 2025. Copyright 2025 by the author(s).

can also be formulated for nonconvex optimization towards *any* objective, matching the range of applicability enjoyed by common models. While only our convex and semiconvex models come with guarantees of convergence to global optimality, all the models we introduce extend the pareto frontier of model size, expressiveness, and optimizability on diverse tasks.

Concretely, we make the following contributions:

- We introduce GA-Planes, a mixture-of-primitives volume parameterization inspired by geometric algebra basis elements. GA-Planes combines any subset of line, plane, and volume features at different resolutions, with an MLP decoder. This GA-Planes family of parameterizations generalizes many existing volume and radiance field models.

- We derive convex and semiconvex reformulations of the GA-Planes training process for certain tasks and a large subset of the GA-Planes model family, to ensure our model optimizes globally regardless of initialization.

- We analyze GA-Planes in the 2D setting and show equivalence to a low-rank plus low-resolution matrix approximation whose expressiveness can be directly controlled by design choices. We demonstrate that this matrix decomposition is expressive for natural images, outperforming the classic low-rank plus sparse approximation.

- We demonstrate convex, semiconvex, and nonconvex GA-Planes' high performance in terms of memory, expressiveness, and optimizability across three volume-fitting tasks: 3D radiance field reconstruction, 3D segmentation, and video segmentation.

## 2. Related Work

**Volume parameterization.** Many volume parameterizations have been proposed and enjoy widespread use in diverse applications. Here we give an overview of representative methods used in computer vision, focusing on methods that parameterize an entire volume (rather than e.g. a surface). These parameterizations exhibit different tradeoffs between memory usage, representation quality, and ease of optimization; richer descriptions are provided in Appendix A.1.

Coordinate MLPs like NeRF (Mildenhall et al., 2020) and Scene Representation Networks (Sitzmann et al., 2019b) are representative of Implicit Neural Representations (INRs), which excel at reducing model size (with decent expressiveness) but suffer from slow optimization. Explicit voxel grid representations like Plenoxels (Sara Fridovich-Keil and Alex

Yu et al., 2022) and Direct Voxel Grid Optimization (Sun et al., 2022) can optimize quickly but require large model size to achieve good expressiveness (resolution). Many other methods (Chen et al., 2022; Fridovich-Keil et al., 2023; Müller et al., 2022; Kerbl et al., 2023; Reiser et al., 2023; Lombardi et al., 2021) find their niche somewhere in between, achieving tractable model size, good expressiveness, and reasonably fast optimization time in exchange for some increased sensitivity (to initialization, randomness, and prior knowledge) in the optimization process. GA-Planes matches or exceeds the performance of strong baselines (Chen et al., 2022; Fridovich-Keil et al., 2023; Barron et al., 2021) in terms of model size and expressiveness, while introducing the option to train by convex or semiconvex optimization with guaranteed convergence to global optimality.

**Radiance field modeling.** Most of the works described above are designed for radiance field modeling, in which the training measurements consist of color photographs from known camera poses. The goal is then to faithfully model the optical density and view-dependent color of light inside a volume so that unseen views can be rendered accurately. Although we do demonstrate superior performance of GA-Planes in this setting, we note that the volumetric rendering formula used in radiance field modeling (Max, 1995; Kajiya, 1986; Mildenhall et al., 2020) yields a nonconvex photometric loss function, regardless of model parameterization.

**3D segmentation.** We test our convex and semiconvex GA-Planes parameterizations on fully convex objectives, namely volume (xyz) segmentation with either indirect 2D tomographic supervision or direct 3D supervision, as well as video (xyt) segmentation. This 3D (xyz) segmentation task has also been studied in recent work (Cen et al., 2023; Uy et al., 2023), though these methods require additional inputs such as a pretrained radiance field model or monocular depth estimator. Our setup is most similar to (Mescheder et al., 2019), which uses an implicit neural representation trained with cross-entropy loss and direct 3D supervision of the occupancy function. Instead of having direct access to this 3D training data, we infer 3D supervision labels via Space Carving (Kutulakos & Seitz, 1999) from 2D image masks obtained by image segmentation (via (Kirillov et al., 2023)).

**Convex neural networks.** Recent work has exposed an equivalence between training a shallow (Pilanci & Ergen, 2020) or deep (Ergen & Pilanci, 2024) neural network and solving a convex program whose structure is defined by the architecture and parameter dimensions of the corresponding neural network. The key idea behind this procedure is to enumerate (or randomly sample from) the possible activation paths through the neural network, and then treat these paths as a fixed dictionary whose coefficients may be optimized according to a convex program.

**Geometric (Clifford) Algebra.** Our model is inspired by Geometric Algebra (GA), which is a powerful framework for modeling geometric primitives (Dorst et al., 2009). The fundamental entity in GA is the multivector, which is a sum of vectors, bivectors, trivectors, etc. In 3D GA, an example is the multivector $\mathbf{e}_1\mathbf{e}_2 + \mathbf{e}_1\mathbf{e}_2\mathbf{e}_3$, representing the sum of a bivector (a plane) and a trivector (a volume). The geometric product in GA allows us to derive a volume element by multiplying a plane and a line, e.g. $(\mathbf{e}_1\mathbf{e}_2)\mathbf{e}_3 = \mathbf{e}_1\mathbf{e}_2\mathbf{e}_3$. We use the shorthand $\mathbf{e}_{123} = \mathbf{e}_1\mathbf{e}_2\mathbf{e}_3$, and similarly for other multivector components throughout. We define the GA-Planes model family to include any volume parameterization that combines any subset (including the complete subset and the empty subset) of the linear geometric primitives $\{\mathbf{e}_1, \mathbf{e}_2, \mathbf{e}_3\}$, planar geometric primitives $\{\mathbf{e}_{12}, \mathbf{e}_{13}, \mathbf{e}_{23}\}$, and/or volumetric primitive $\{\mathbf{e}_{123}\}$ into a trivector, with a (potentially convexified) MLP feature decoder. While our approach leverages the linear algebraic foundations of GA to construct feature grids, it does not fully exploit the entire range of GA operations and avoids the computationally heavy machinery typically associated with GA. Previous works have integrated GA multiplications and equivariance properties into NN layers (Brandstetter et al., 2022; Ruhe et al., 2023a;b), however, this work is the first to use GA in neural volume models to our knowledge.

## 3. Model

### 3.1. The GA-Planes Model Family

A GA-Planes model represents a volume using a combination of geometric algebra features $\mathbf{e}_c$ derived by interpolating the following parameter grids:

- Line (1-dimensional) feature grids $\{\mathbf{g}_1, \mathbf{g}_2, \mathbf{g}_3\}$, where each grid has shape $[r_1, d_1]$ with spatial resolution $r_1$ and feature dimension $d_1$.

- Plane (2-dimensional) feature grids $\{\mathbf{g}_{12}, \mathbf{g}_{13}, \mathbf{g}_{23}\}$, where each grid has shape $[r_2, r_2, d_2]$ with spatial resolution $r_2$ and feature dimension $d_2$.

- A single volume feature grid $\{\mathbf{g}_{123}\}$ with shape $[r_3, r_3, r_3, d_3]$.

A GA-Planes model may include copies of a given basis element with different resolution and feature dimensions, to effectively capture multi-resolution signal content. The x, y, and z spatial resolutions of each grid may also differ in practice; for notational simplicity we write isotropic resolutions.

We first extract features corresponding to $q = (x, y, z) \in \mathbb{R}^3$ from each of our line, plane, and volume feature grids $\mathbf{g}_c$ by linear, bilinear, and trilinear interpolation, respectively:

$$\mathbf{e}_c := \psi\big(\mathbf{g}_c, \pi_c(q)\big), \tag{1}$$

where $\pi_c$ projects $q$ onto the coordinates of the $c$'th feature grid $\mathbf{g}_c$ and $\psi$ denotes (bi/tri)linear interpolation. The resulting feature $\mathbf{e}_c$ is a vector of length $d_1$ if $c \in \{1, 2, 3\}$, $d_2$ if $c \in \{12, 13, 23\}$ or $d_3$ if $c = 123$. We repeat this projection and interpolation procedure over each $\mathbf{g}_c$, and combine the resulting feature vectors by any combination of elementwise multiplication ($\circ$), addition ($+$), and concatenation ($\odot$) along the feature dimension. Finally, the combined feature vector is decoded using an MLP decoder D. The decoder can take as input both the feature vector arising from the feature grids as well as possible auxiliary inputs, such as (positionally encoded) viewing direction. We consider any model that fits the above description to fall into the GA-Planes family. The specific models we use for nonconvex, semiconvex, and convex optimization are illustrated in Figure 1.

Our experiments focus primarily on two specific GA-Planes models that exemplify some of the strongest convex and nonconvex representations in the GA-Planes family. For our experiments including convex optimization, namely 3D segmentation with 2D or 3D supervision, and video segmentation, we use the following GA-Planes model (illustrated in the second and third rows of Figure 1) which can be trained by either convex, semiconvex, or nonconvex optimization as described in the following subsections:

$$\text{D}(\mathbf{e}_1 \odot \mathbf{e}_2 \odot \mathbf{e}_3 \odot \mathbf{e}_{12} \odot \mathbf{e}_{13} \odot \mathbf{e}_{23} \odot \mathbf{e}_{123}). \tag{2}$$

Here we use $\odot$ to denote concatenation of features. For our radiance field experiments, since the objective function is inherently nonconvex, we use the following nonconvex member of the GA-Planes family (illustrated with multiresolution feature grids in the first row of Figure 1):

$$\text{D}((\mathbf{e}_1 \circ \mathbf{e}_2 \circ \mathbf{e}_3) \odot (\mathbf{e}_1 \circ \mathbf{e}_{23}) \odot (\mathbf{e}_2 \circ \mathbf{e}_{13}) \odot (\mathbf{e}_3 \circ \mathbf{e}_{12}) \odot \mathbf{e}_{123}), \tag{3}$$

which leverages geometric algebra to multiply ($\circ$) lower-dimensional (vector and bivector) features together into 3D volume (trivector) features, but cannot be convexified because of this multiplication. We use multi-resolution copies of the line and plane feature grids $\mathbf{g}_1, \mathbf{g}_2, \mathbf{g}_3, \mathbf{g}_{12}, \mathbf{g}_{13}, \mathbf{g}_{23}$, but only a single resolution for the volume grid $\mathbf{g}_{123}$ since it is already at lower resolution. For our nonconvex experiments the decoder D is a standard fully-connected ReLU neural network; decoder details for our semiconvex and convex models are presented in the following subsections.

### 3.2. Semiconvex GA-Planes

For our segmentation experiments (with volumes and videos), we use the GA-Planes architecture in Equation (2), with concatenation instead of multiplication of features; we denote this concatenated feature vector as $f(q)$, the input to the decoder. Our semiconvex formulation of this model

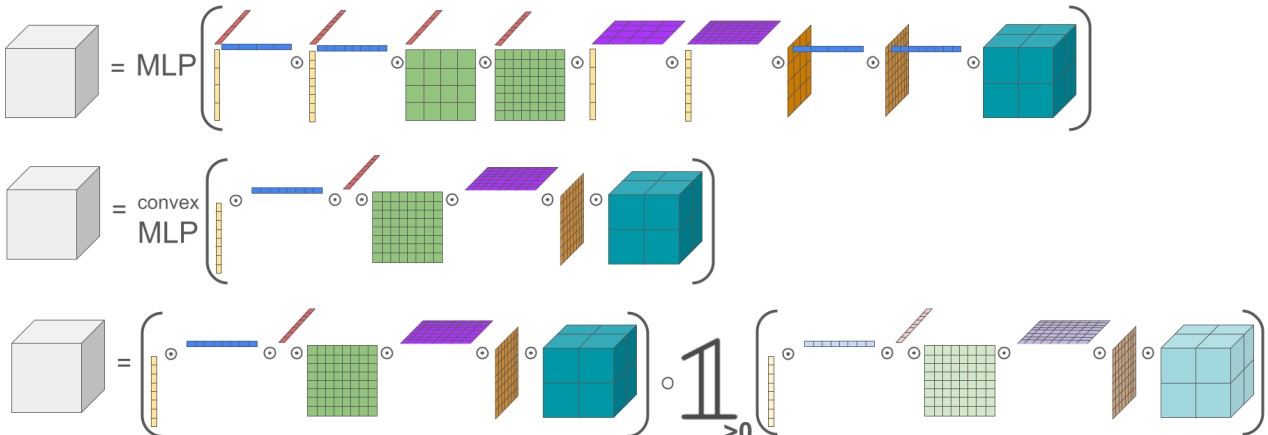

**Figure 1:** Overview of the GA-Planes models we use in our experiments. Our nonconvex model (top) uses a standard MLP decoder and multiplication of features when the result yields a volume under geometric algebra; it also concatenates features across multi-resolution grids. Our semiconvex (middle) and convex (bottom) models use a single resolution for each feature grid, and avoid multiplication of features since that would induce nonconvexity. The pastel-colored grids inside the indicator function of the convex model are frozen at initialization and used as fixed ReLU gating patterns. $\odot$ denotes concatenation and $\circ$ denotes elementwise multiplication.

uses a convex MLP (Pilanci & Ergen, 2020) as the decoder:

$$\tilde{y}(q) = \sum_{i=1}^{h} (W_i^\top f(q)) \mathbb{1}[\overline{W}_i^\top f(q) \geq 0]. \quad (4)$$

Here $W$ denotes the trainable hidden layer MLP weights, and $\overline{W}$ denotes the same weights frozen at initialization inside the indicator function. The indicator function, denoted as $\mathbb{1}[*]$, returns 1 if the argument is true, and 0 otherwise. The indicator function here serves as a random gating pattern that takes the place of the ReLU in a standard nonconvex MLP, where the gating pattern would be optimized rather than fixed in a random pattern. Although this MLP decoder is fully convex, we refer to this model as semiconvex (in particular biconvex; see (Sahiner et al., 2024)) because the combined grid features $f(q)$ are multiplied by the trainable MLP hidden layer weights $W$, though the objective is separately convex in each of these parameters.

### 3.3. Convex GA-Planes

For our segmentation experiments (with volumes and videos), we also present a fully convex GA-Planes model that is similar to the semiconvex model described above, except that we fuse the learnable weights of the MLP decoder with the weights of the feature mapping, to remove the product of parameters (which is semiconvex but not convex). Our convex model is:

$$\tilde{y}(q) = \sum_{c \in \{1,2,3,12,13,23,123\}} \mathbb{1}_{d(c)}^\top (\mathbf{e}_c \circ \mathbb{1}[\overline{\mathbf{e}}_c \geq 0]), \quad (5)$$

where the features $\mathbf{e}_c$ are interpolated from optimizable parameter grids with feature dimension $d(c) \in \{d_1, d_2, d_3\}$, whereas the gating variables $\overline{\mathbf{e}}_c$ inside the indicator function are derived from the same grids frozen at their initialization values to preserve convexity. Here $\circ$ denotes elementwise product of vectors. These indicator functions take the same role as the ReLU in a nonconvex MLP, using a sampling of random activation patterns based on the grid values at initialization.

## 4. Theory

### 4.1. Equivalence to Matrix Completion in 2D

In three dimensions, the complete set of geometric algebra feature grids are those that we include in the GA-Planes family: $\{\mathbf{g}_1, \mathbf{g}_2, \mathbf{g}_3, \mathbf{g}_{12}, \mathbf{g}_{13}, \mathbf{g}_{23}, \mathbf{g}_{123}\}$. In two dimensions, this reduces to: $\{\mathbf{g}_1, \mathbf{g}_2, \mathbf{g}_{12}\}$. In this 2D setting, we can analyze different members of our GA-Planes family and show equivalence to various formulations of the classic matrix completion problem.

**Notation.** As usual, we use $\circ$ to denote elementwise multiplication and $\odot$ to denote concatenation. We use $\mathbb{1}_{a \times b}$ to denote the all-ones matrix of size $a \times b$ and $\mathbb{1}[\cdot]$ to denote the indicator function, which evaluates to 1 when its argument is positive and 0 otherwise. Our theorem statements consider equivalence to a matrix completion problem with target matrix $M \in \mathbb{R}^{m \times n}$ and low-rank components $U \in \mathbb{R}^{m \times k}, V \in \mathbb{R}^{n \times k}$ to be optimized. We include theoretical results for 2D GA-Planes models that combine features by addition ($+$), multiplication ($\circ$), or concatena-

tion ($\odot$) and decode features using a linear decoder (as a warmup), a convex MLP, or a nonconvex MLP. The most illuminating results are presented in the theorem statements that follow; the rest (and all proofs) are deferred to Appendix A.2.

**Assumptions.** Our theorem statements assume that the line feature grids have the same spatial resolution as the target matrix, and thus do not specify the type of interpolation. However, the results hold even if the dimensions do not match and nearest neighbor interpolation is used; the empirical performance is similar or even slightly improved in practice by using (bi)linear interpolation of features (see Appendix A.3 and A.2 for a discussion of other interpolation methods). The theorems assume that the optimization objective is to minimize the Frobenius norm of the error matrix; this is equivalent to minimizing mean squared error measured directly in the representation space. This objective function is the one we use for our convex experiments (video and volume segmentation fitting), where we have access to direct supervision; our radiance field experiments instead use indirect measurements (along rays) that are not equivalent to the setting of the theorems.

**Theorem 1.** *The two-dimensional representation* $D(\mathbf{e}_1 + \mathbf{e}_2)$ *with linear decoder* $D(f(q)) = \alpha^T f(q)$ *is equivalent to a low-rank matrix completion model with the following structure:*

$$\min_{U,V} \|M - (U\mathbb{1}_{k \times n} + \mathbb{1}_{m \times k}V^T)\|_F^2. \quad (6)$$

*These two models are equivalent in the sense that* $U^* = \mathbf{g}_1^* diag(\alpha^*)$ *and* $V^* = \mathbf{g}_2^* diag(\alpha^*)$ *where* $U^*, V^*$ *is the optimal solution to the low-rank matrix completion problem in Equation* (6) *and* $\mathbf{g}_1^*, \mathbf{g}_2^*, \alpha^*$ *are the optimal grid features and linear decoder for the* $D(\mathbf{e}_1 + \mathbf{e}_2)$ *model.*

*The two-dimensional representation* $D(\mathbf{e}_1 \circ \mathbf{e}_2)$ *with the same linear decoder is equivalent to the standard low-rank matrix completion model:*

$$\min_{U,V} \|M - UV^T\|_F^2. \quad (7)$$

*These two models are equivalent in the same sense as above, except that* $V^* = \mathbf{g}_2^*$.

**Remark.** Using a linear decoder reveals a dramatic difference in representation capacity between feature addition (or concatenation) and multiplication. Using addition, the maximum rank of the matrix approximation is 2 regardless of the feature dimension $k$. Using multiplication, the maximum rank of the approximation is $k$. With feature multiplication, the optimal values of the feature grids are identical to the rank-thresholded singular value decomposition (SVD) of $M$, where the feature grids $\mathbf{g}_1$ and $\mathbf{g}_2$ recover the left and

right singular vectors and the decoder $\alpha$ learns the singular values of $M$. This is the optimal rank $k$ approximation of a matrix $M$.

**Theorem 2.** *The two-dimensional representation* $D(\mathbf{e}_1 + \mathbf{e}_2 + \mathbf{e}_{12})$ *with linear decoder* $D(f(q)) = \alpha^T f(q)$ *is equivalent to a low-rank plus low-resolution matrix completion model with the following structure:*

$$\min_{U,V,L} \|M - (U\mathbb{1}_{k \times n} + \mathbb{1}_{m \times k}V^T + \varphi(L))\|_F^2, \quad (8)$$

*where* $L \in \mathbb{R}^{m_l \times n_l}$ *is the low-resolution component to be learned, with upsampling (interpolation) function* $\varphi$. *These two models are equivalent in the sense that* $U^* = \mathbf{g}_1^* diag(\alpha^*)$, $V^* = \mathbf{g}_2^* diag(\alpha^*)$, *and* $L^* = \mathbf{g}_{12}^* \alpha^*$, *where* $U^*, V^*, L^*$ *is the optimal solution to the low-rank plus low-resolution matrix completion problem in Equation* (8) *and* $\mathbf{g}_1^*, \mathbf{g}_2^*, \mathbf{g}_{12}^*, \alpha^*$ *are the optimal grid features and linear decoder for the* $D(\mathbf{e}_1 + \mathbf{e}_2 + \mathbf{e}_{12})$ *model.*

*The two-dimensional representation* $D(\mathbf{e}_1 \circ \mathbf{e}_2 + \mathbf{e}_{12})$ *with the same linear decoder is equivalent to a low-rank plus low-resolution matrix completion model:*

$$\min_{U,V,L} \|M - (UV^T + \varphi(L))\|_F^2. \quad (9)$$

*These two models are equivalent in the same sense as above, except that* $V^* = \mathbf{g}_2^*$.

**Remark.** Theorem 2 describes the behavior of a 2D, linear-decoder version of our GA-Planes model, both the version with addition/concatenation of features (Equation (2)) and the version with multiplication of features (Equation (3)). Extending the same idea to 3D, we can interpret GA-Planes as a low-rank plus low-resolution approximation of a 3D tensor (volume). We can understand this model as first fitting a low-resolution volume and then finding a low-rank approximation to the high-frequency residual volume. When we use multiplication of features, the low-rank residual approximation is optimal and analogous to the rank-thresholded SVD.

**Theorem 3.** *The two-dimensional representation* $D(\mathbf{e}_1 \circ \mathbf{e}_2)$ *with a two-layer convex MLP decoder* $D(f(q)) = \sum_{i=1}^{h} (W_i^\top f(q)) \mathbb{1}[\overline{W}_i^\top f(q) \geq 0]$ *is equivalent to a masked low-rank matrix completion model:*

$$\min_{U,V,W} \left\|M - \sum_{i,j} W_{i,j} U_j V_j^\top \circ B_i\right\|_F^2, \quad (10)$$

*where* $W \in \mathbb{R}^{h \times k}$ *contains the trainable weights of the convex MLP decoder, with indices* $j = 1, \ldots, k$ *for the input dimension and* $i = 1, \ldots, h$ *for the hidden layer dimension.* $B_i \in \mathbb{R}^{m \times n}$ *denotes the binary masking matrix formed by random, fixed gates of the convex MLP decoder;* $B_i = \mathbb{1}[\sum_j \overline{W}_{i,j} U_j V_j^\top \geq 0]$, *where* $\overline{W}$ *denotes the weight matrix* $W$ *with values fixed at random initialization.*

This matrix completion model and our GA-Planes model $D(\mathbf{e}_1 \circ \mathbf{e}_2)$ with convex MLP decoder are equivalent in the sense that $U^* = \mathbf{g}_1^*$, $V^* = \mathbf{g}_2^*$, and $W^* = W^*$, where $U^*, V^*, W^*$ is the optimal solution to the masked low-rank matrix completion problem Equation (10) and $\mathbf{g}_1^*, \mathbf{g}_2^*, W^*$ are the optimal grid features and convex MLP decoder weights for the $D(\mathbf{e}_1 \circ \mathbf{e}_2)$ model. The optimal mask matrices $B_i^*$ are defined by the fixed random weight initialization $\overline{W}$ and the optimal singular vector matrices $U^*, V^*$.

**Remark.** We can interpret the matrix completion model of Equation (10) as a sum of $h$ different low-rank approximations, where the matrices within each of the $h$ groups are constrained to share the same singular vectors $U_j, V_j$. The binary masks $B_i$ effectively allow each of these $h$ low-rank approximations to attend to (or complete) a different part of the matrix $M$ before being linearly combined through the weights (singular values) $W_{i,j}$. The upper limit of the rank of this matrix approximation is thus $\min(n, m)$, because the mask matrices can arbitrarily increase the rank beyond the constraint faced by models with a linear decoder. Note that if the feature grids $\mathbf{g}_1$ and $\mathbf{g}_2$ have spatial resolution $r_1$ less than $\min(n, m)$, the maximum rank will be $r_1$.

**Theorem 4.** *The two-dimensional representation $D(\mathbf{e}_1 \circ \mathbf{e}_2)$ with a standard two-layer MLP decoder $D(f(q)) = \alpha^T (W f(q))_+$ is equivalent to a low-rank matrix completion model with the following structure:*

$$\min_{U,V,W,\alpha} \left\| M - \sum_{i=1}^{h} \alpha_i \Big( \sum_{j=1}^{k} W_{i,j} U_j V_j^\top \Big)_+ \right\|_F^2, \quad (11)$$

*where $W \in \mathbb{R}^{h \times k}$ is the weight matrix for the MLP decoder's hidden layer (with width $h$) and $\alpha \in \mathbb{R}^h$ is the weight vector of the MLP decoder's output layer.*

This matrix completion model and our GA-Planes model $D(\mathbf{e}_1 \circ \mathbf{e}_2)$ with nonconvex MLP decoder are equivalent in the sense that $U^* = \mathbf{g}_1^*$, $V^* = \mathbf{g}_2^*$, $W^* = W^*$, and $\alpha^* = \alpha^*$, where $U^*, V^*, W^*, \alpha^*$ is the optimal solution to the masked low-rank matrix completion problem Equation (11) and $\mathbf{g}_1^*, \mathbf{g}_2^*, W^*, \alpha^*$ are the optimal grid features and MLP decoder for the $D(\mathbf{e}_1 \circ \mathbf{e}_2)$ model.

**Remark.** The upper limit of the rank of this matrix approximation is $\min(n, m, r_1)$, the same as with a convex MLP decoder.

We summarize the maximum attainable ranks of different 2D models in Table 1 (see Appendix A.2.3 and A.2.4 for matrix representations of $D(\mathbf{e}_1 + \mathbf{e}_2)$ and $D(\mathbf{e}_1 \odot \mathbf{e}_2)$ with convex and nonconvex MLP decoders). Experimental validation of these theoretical results on the task of 2D image compression is provided with a comparison of interpolation schemes in Figure 7 in the appendix.

| Model | Linear decoder | convex MLP decoder | MLP decoder |
|---|---|---|---|
| $D(\mathbf{e}_1 + \mathbf{e}_2)$ | 2 | $r_1$ | $r_1$ |
| $D(\mathbf{e}_1 \odot \mathbf{e}_2)$ | 2 | $r_1$ | $r_1$ |
| $D(\mathbf{e}_1 \circ \mathbf{e}_2)$ | $k$ | $r_1$ | $r_1$ |

**Table 1:** Maximum attainable ranks of different 2D GA-Planes models, using only line features. Here $k$ is the feature dimension and $r_1$ is the spatial dimension of the features, which need never exceed $\min(m, n)$. Replacing a linear decoder with a convex or nonconvex MLP can dramatically increase the rank of the representation.

### 4.2. Practical Implications

The upper bounds on model expressivity summarized in Table 1 inform practitioners that (1) If the decoder is linear (or more generally, simple), features should be combined by multiplication rather than addition or concatenation, and (2) If an MLP decoder is used, the rank of the representation will be limited by the resolution of the grid features rather than by their feature dimension. Indeed this second theoretical insight is practically useful in finding the set of parameter allocations that work best in our radiance field experiments (see Figure 3 and Table 5), in which we prioritize maintaining some features with high resolution even at the cost of reducing the feature dimension.

The upper bounds in Table 1 are also complemented by 2D image fitting experiments in Figure 7, which directly inform practice in 3 ways beyond the insights from Table 1: (1) combining features by multiplication rather than addition is beneficial even when using an MLP decoder, (2) a standard nonconvex MLP decoder outperforms a convex MLP decoder, even though both representations have the same maximum rank, and (3) continuous interpolation of features outperforms nearest neighbor interpolation of features.

### 4.3. Interpretation: Low Rank + Low Resolution

Combining multiple parameterization strategies with complementary representation capacities is a time-honored strategy in signal processing. A classic example is the combination of sparse and low-rank models used to represent matrices in the compressive sensing literature (Chandrasekaran et al., 2009). As shown in Theorem 2, we can view the GA-Planes family as following a similar strategy with a combination of low-rank and low-resolution approximations. This low-rank plus low-resolution parameterization is generally easier to train, because sparse models must either store large numbers of empty values (high memory) or store the locations of nonzero entries and suffer from poorly-conditioned spatial gradients (difficult optimization). Indeed there are volume parameterizations that utilize sparsity, such as point clouds, surface meshes, sparse grids, surfels, and Gaussian splats, but these tend to be more challenging to optimize

(e.g. requiring high memory (Sara Fridovich-Keil and Alex Yu et al., 2022) or heuristics and good initialization (Kerbl et al., 2023)).

We illustrate this low-rank plus low-resolution interpretation in Figure 2 with a simple experiment, in which we approximate a grayscale image (the *astronaut* image from SciPy) using either a sum of low-rank and low-resolution components (similar to the structure of GA-Planes) or the classic sum of low-rank and sparse components. In this experiment we compute the optimal low-rank components of each model type using the SVD, which corresponds to multiplication of features.

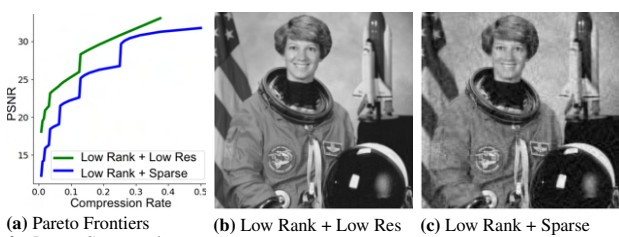

**(a)** Pareto Frontiers for Image Compression  
**(b)** Low Rank + Low Res PSNR 29.60  
**(c)** Low Rank + Sparse PSNR 26.26

**Figure 2:** For a natural image, approximation as a sum of low rank and low resolution components (green curve and subfigure b) achieves higher fidelity compared to the classic matrix decomposition as a sum of low rank and sparse components (blue curve and subfigure c), with the same parameter budget (18.75% of the original image size, for subfigures b and c). The GA-Planes model family generalizes the idea of a low rank plus low resolution approximation to three dimensions.

## 5. Experiments

### 5.1. Radiance Field Modeling

Our experiments for the radiance field reconstruction task are built with NeRFStudio (Tancik et al., 2023) and use all 8 scenes from NeRF-Blender (Mildenhall et al., 2020). We train each volume representation based on the standard photometric loss (Max, 1995; Kajiya, 1986; Mildenhall et al., 2020). Our results are summarized in Figure 3, which reports PSNR, SSIM (Wang et al., 2004), and LPIPS (Zhang et al., 2018) for each model as a function of its size. We provide renderings and per-scene pareto-optimal curves in Figure 4 and A.5. We also compare memory and training time requirements in A.7.

Because the photometric loss function is inherently nonconvex, we use our most expressive GA-Planes parameterization as defined in Equation (3). We compare this with several popular models as implemented in NeRFStudio: Mip-NeRF (Barron et al., 2021), TensoRF (Chen et al., 2022), and K-Planes (Fridovich-Keil et al., 2023). For K-Planes, we include versions with and without proposal

sampling, a strategy for efficiently allocating ray samples. Proposal sampling is the default for K-Planes, but we include a version without proposal sampling because none of the other models in this experiment use proposal sampling. We also include several ablations of our GA-Planes model: one with only the volume features (similar to a multiresolution version of DVGO (Sun et al., 2022) or Plenoxels (Sara Fridovich-Keil and Alex Yu et al., 2022)), one with only the line features (similar to a multiresolution TensoRF-CP), and one with only the line-plane products (similar to a multiresolution TensoRF-VM (Chen et al., 2022)). At large model sizes (∼10 million parameters) most models including GA-Planes perform well. As model size shrinks, only GA-Planes and its line-only ablation reach comparable metrics as the larger models. Detailed descriptions of parameter allocations for each model are provided in Appendix A.6.

### 5.2. 3D Segmentation

Our experiments for 3D segmentation use opacity masks from the NeRF-Blender *lego* scene (Mildenhall et al., 2020). We "lift" these 2D segmentation masks to 3D. We compare the linear and nonlinear feature combination versions of GA-Planes and Tri-Planes (Chan et al., 2022; Fridovich-Keil et al., 2023) (models with plane features only).

**2D Supervision.** In this experiment we use 2D tomographic supervision, in which we minimize the mean squared error between the ground truth 2D object segmentation masks and the average ray density at each viewpoint. In Table 2 we report the intersection over union (IOU) metric on thresholded projections from our trained model on test views. These results validate that the GA-Planes architecture in Equation (2) works well regardless of convex, semiconvex, or nonconvex formulation–whereas the Tri-Planes model requires a nonconvex decoder for good performance. Note that Tri-Planes with feature concatenation performs on-par with planes with addition, thus, we keep the original formulation with addition for our comparisons.

**3D Supervision.** Our second set of 3D segmentation experiments leverages direct 3D supervision via Space Carving (Kutulakos & Seitz, 1999), using the principle that if any ray passing through a given 3D coordinate is transparent, the density at that 3D coordinate must be zero. Our results with 3D supervision, in Table 2, parallel those with 2D supervision: GA-Planes performs well regardless of convex, semiconvex, or nonconvex formulation, whereas the Tri-Planes model performs decently with a nonconvex decoder but much worse with convex or semiconvex formulation.

### 5.3. Video Segmentation

Our video segmentation task is similar to volume segmentation with 3D supervision: here the dimensions are $x, y, t$ rather than $x, y, z$, and the supervision is performed directly

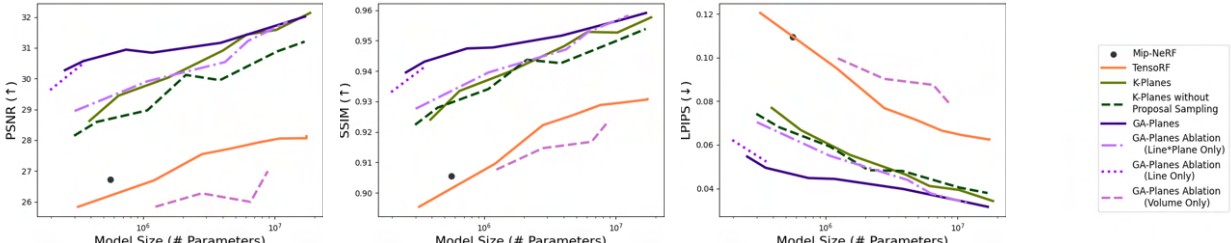

**Figure 3:** Results on radiance field reconstruction. Nonconvex GA-Planes (with feature multiplication) offers the most efficient representation: when the model is large it performs comparably to the state of the art models, but when model size is reduced it retains higher performance than other models. Here all models are trained for the same number of epochs on all 8 scenes from the Blender dataset, and the average results are shown.

| | Task 1 (2D Tomographic) | | | Task 2 (3D Space Carving) | | | Task 3 (Video Fitting) | | |
|---|---|---|---|---|---|---|---|---|---|
| | Convex | Semiconvex | Nonconvex | Convex | Semiconvex | Nonconvex | Convex | Semiconvex | Nonconvex |
| GA-Planes (with ∘) | - | - | 0.877 | - | - | 0.926 | - | - | 0.974 |
| GA-Planes (with ⊙) | 0.875 | 0.883 | 0.880 | 0.932 | 0.957 | 0.964 | 0.913 | 0.975 | 0.981 |
| Tri-Planes (planes with ∘, like K-Planes) | - | - | 0.877 | - | - | 0.881 | - | - | 0.727 |
| Tri-Planes (planes with +, like (Chan et al., 2022)) | 0.681 | 0.868 | 0.863 | 0.642 | 0.636 | 0.941 | 0.557 | 0.647 | 0.732 |

**Table 2:** Tasks 1&2: Intersection over union (IOU) for recovering novel view object segmentation masks across two different experiments: Experiment 1 involves segmentation mask training with 2D tomographic supervision, and Experiment 2 involves segmentation mask training with 3D Space Carving supervision. Task 3: Intersection over union (IOU) for temporal superresolution of segmentation masks in a video, computed on held-out test frames. Models that involve multiplication of features can only be trained by nonconvex optimization.

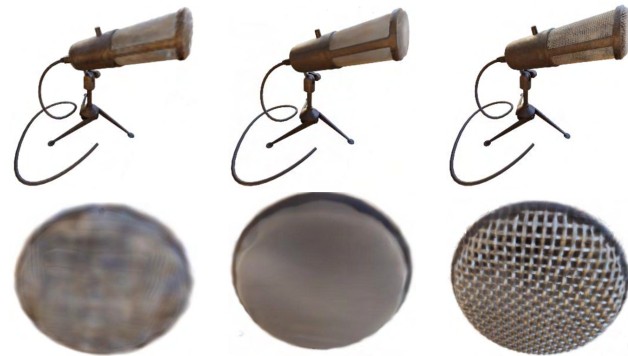

**Figure 4:** Rendering comparison for the *mic* scene: TensoRF on the left, K-Planes in the middle, GA-Planes on the right (0.32, 0.39, 0.25 M parameters, respectively). Visual results on other scenes are in Appendix A.6.

in 3D using segmentation masks for a subset of the video frames (every third frame is held out for testing). Our dataset preparation pipeline uses the skateboarding video and pre-processing steps described at Labelbox.com, which involves first extracting a bounding box with YOLOv8 (Jocher et al., 2023) and then segmenting the skateboarder with SAM (Kirillov et al., 2023). This is essentially a temporal super-resolution task on segmentation masks.

Our results are summarized in Figure 5 and Table 2. We find that GA-Planes performs well across convex, semiconvex, and nonconvex formulations, though its performance

is slightly reduced under fully convex training, perhaps because the convex model size is slightly reduced due to fusing the decoder parameters into the feature grids. In contrast, the simpler Tri-Plane models perform poorly on this task regardless of training strategy: they fail to learn the temporal sequence of the video, producing masks that focus only on the skateboarder's less-mobile core.

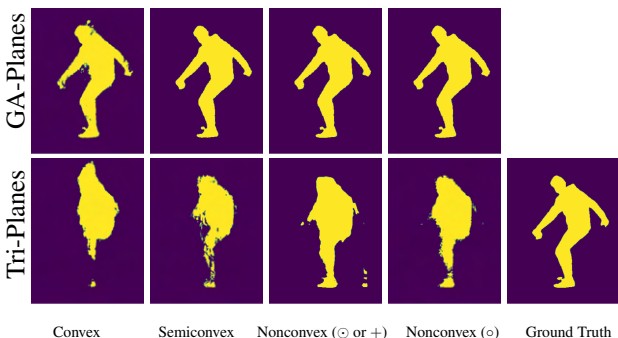

**Figure 5:** Intersection over union (IOU) for predicting segmentation masks for unseen frames within a video of a segmented skateboarder.

### 5.4. Benefits of Convexity

In most of our experiments, all models (convex, semiconvex, and nonconvex) are large enough that they are able to optimize well. However, we highlight a benefit of our convex and semiconvex models that they enjoy more stable opti-

mization even with very small model sizes. In Figure 6 we compare test intersection-over-union (IoU) curves for very small models for our video fitting task (hidden dimension 4 in the decoder MLP, and feature dimensions $[d_1, d_2, d_3] = [4, 4, 2]$ and resolutions $[r_1, r_2, r_3] = [32, 32, 16]$ for line, plane, and volume features, respectively). We repeat optimization with 10 different random seeds used to initialize the optimizable parameters (gating weights for the convex and semiconvex models are fixed) and report the mean and standard deviation for each model's final IoU in Table 3. While the convex and semiconvex models enjoy stable training curves across random seeds, we find that the nonconvex model experiences much more volatile training behavior (completely failing to optimize with some of the random seeds).

On average, the semiconvex model performs best, followed by the convex model, with the nonconvex model performing worst on average. We can also see from the standard deviation values that the convex model is extremely stable with respect to random initialization, as is the semiconvex model. In contrast, the nonconvex model is highly unstable, with some seeds failing to converge at all.

|  | Mean | Standard Deviation |
|---|---|---|
| Convex GA-Planes | 0.639 | 0.000 |
| Semiconvex GA-Planes | 0.716 | 0.022 |
| Nonconvex GA-Planes | 0.567 | 0.229 |

**Table 3:** Mean and standard deviation of intersection over union (IOU) scores for video segmentation over 10 seeds, with a small GA-Planes model. Figure 6 visualizes these results over the course of training.

## 6. Discussion

In this work we introduce GA-Planes, a family of volume parameterizations that generalizes many existing representations (see Appendix A.1). We specifically focus on two members of the GA-Planes family (with concatenation versus multiplication of features), and offer both theoretical interpretation and empirical evaluation of these models. Our nonconvex GA-Planes model shows pareto-optimal performance in terms of model size and quality on fitting a 3D radiance field, while our convex and semiconvex GA-Planes formulations are effective on several 3D linear inverse problems. In 2D, we show connections between GA-Planes and a low-rank plus low-resolution matrix completion model, and derive how this model's rank capacity is affected by various design decisions.

**Limitations.** Here we focus on 3D (or smaller) representations, rather than higher dimensions (e.g. dynamic volumes), and we demonstrate GA-Planes for reconstruction rather than generation tasks. Both of these extensions are

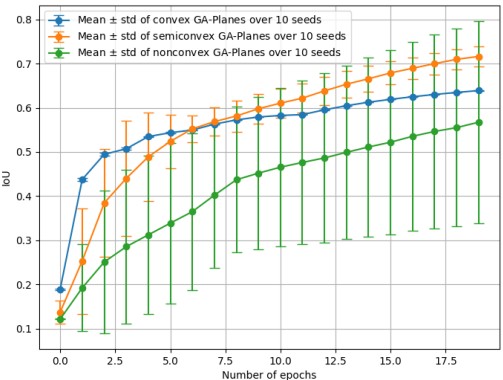

**Figure 6:** Test performance throughout training on our video fitting task, for a very small GA-Planes model across 10 random seeds. We find that the favorable optimization landscape of our convex and semiconvex models enables reliable training across seeds, whereas the nonconvex model fails to fit any test frame with some of the seeds.

promising avenues for extending GA-Planes. In our experiments, we use the same first-order optimization algorithm for all models. However, our convex GA-Planes formulation is compatible with any convex solver (e.g. cvxpy), and we expect its performance may improve by leveraging these efficient convex optimization algorithms. We demonstrate preliminary benefits of convexity and semiconvexity in terms of training stability in Section 5.4.

## Impact Statement

This paper presents work whose goal is to advance the field of Machine Learning. We specifically focus on efficiency of representations and optimization. There are many potential societal consequences of our work, none which we feel must be specifically highlighted here.

## Acknowledgements

We are grateful to Axel Levy for helpful discussions on space carving, and to Abhishek Shetty for helpful discussions on matrix completion. This work was supported in part by the NSF Mathematical Sciences Postdoctoral Research Fellowship under award number 2303178, in part by the National Science Foundation (NSF) under Grant DMS-2134248, in part by the NSF CAREER Award under Grant CCF-2236829, in part by the U.S. Army Research Office Early Career Award under Grant W911NF-21-1-0242, and in part by the Office of Naval Research under Grant N00014-24-1-2164.

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

# A. Appendix

## A.1. Context for GA-Planes

Table 4 summarizes some representative volume models popular in computer vision, and how they relate to GA-Planes along the three-way pareto frontier of model size, expressiveness, and optimizability. In Figure **??**, we provide an overview of our GA-Planes models.

*Implicit Neural Representations* (INRs) or Coordinate Neural Networks (Mildenhall et al., 2020; Sitzmann et al., 2019b; Tancik et al., 2020; Sitzmann et al., 2020; Saragadam et al., 2023) parameterize the volume implicitly through the weights of a neural network, typically a multilayer perceptron (MLP) with some modification to overcome spectral bias and represent high frequency content. These models tend to provide decent expressiveness with very small model size; their main drawback is slow optimization.

*Voxel grids* (Kutulakos & Seitz, 1999; Sara Fridovich-Keil and Alex Yu et al., 2022; Sun et al., 2022; Sitzmann et al., 2019a) are perhaps the most traditional parameterization of a volume, where each parameter denotes the function value (density, color, a latent feature, etc.) at a specific grid cell location within the volume. These voxel values can then be combined into a continuous function over the 3D space by some form of interpolation, following the standard Nyquist sampling and reconstruction paradigm of digital signal processing (Oppenheim, 1999). Voxels offer direct control over expressivity (via resolution) and are easily optimized; their main drawback is memory usage because the number of parameters grows cubically with the spatial resolution.

*Tensor factorizations* (Chen et al., 2022; Chan et al., 2022; Fridovich-Keil et al., 2023) parameterize a 3D volume as

|  | Model Size | Expressiveness | Optimizability |
|---|---|---|---|
| Coordinate MLP (NeRF, SRN) | ✓ | ∼ | ✗ |
| Voxels (Space Carving, Plenoxels, DVGO) | ✗ | ✓ | ✓ |
| Tensor Factorization (TensoRF, K-Planes) | ∼ | ∼ | ∼ |
| Hash Embedding (Instant-NGP) | ∼ | ✓ | ∼ |
| Point Cloud / Splat (3D Gaussian Splatting) | ∼ | ✓ | ∼ |
| Mixture of Primitives (MVP, MERF) | ∼ | ✓ | ∼ |
| GA-Planes (Nonconvex) | ∼ | ✓ | ∼ |
| GA-Planes (Convex) | ∼ | ✓ | ✓ |
| GA-Planes (Semi-Convex) | ∼ | ✓ | ✓ |

**Table 4: Context.** All volume models face a tradeoff between memory efficiency, expressiveness, and optimizability. The qualitative categorizations here are based on the tradeoffs achieved by representative example methods listed in each category. *Model Size* denotes memory usage during training; other methods exist to compress trained models, e.g. for rendering on mobile hardware. *Optimizability* denotes both speed and stability of optimization/training. For example, Coordinate MLPs tend to train slowly, while Splats train quickly but are sensitive to initialization.

a combination of lower-dimension objects, namely vectors and matrices (lines and planes). Tensor factorizations tend to balance the three attributes somewhat evenly, offering decent expressiveness and optimizability while using more memory than an INR but less than a high resolution voxel grid.

*Hash embeddings* (Müller et al., 2022; Tancik et al., 2023) are similar to voxels, but replace the explicit voxel grid in 3D with a multiresolution 3D hash function followed by a small MLP decoder to disambiguate hash collisions. They can optimize very quickly and with better memory efficiency compared to voxels; quality is mixed with good high-resolution details but also some high-frequency noise likely arising from unresolved hash collisions or sensitivity to random initialization.

*Point clouds / splats* (Kerbl et al., 2023; Schönberger & Frahm, 2016; Schönberger et al., 2016) represent a volume as a collection of 3D points or blobs, where the points need not be arranged on a regular grid. They are highly expressive and less memory-intensive than voxels (but still more so than some other methods). They can optimize very quickly but often require heuristic or discrete optimization strategies that result in sensitivity to initialization.

*Mixture of primitives* (Reiser et al., 2023; Lombardi et al., 2021) models combine multiple of the above representation strategies to balance their strengths and weaknesses. For example, combining low resolution voxels with a high resolution tensor factorization is an effective strategy to improve on the expressiveness of tensor factorizations without resorting to the cubic memory requirement of a high resolution voxel grid; this strategy underlies both MERF (Reiser et al., 2023) and GA-Planes.

We emphasize that all of these existing methods (except perhaps voxels) require nonconvex optimization, often for a feature decoder MLP, and thus risk getting stuck in suboptimal local minima depending on the randomness of initialization and the trajectory of stochastic gradients. In practice, as described above, some of the prior methods exhibit greater optimization stability than others, though none (except voxels in limited settings) come with guarantees of convergence to global optimality. In contrast, both the convex and semiconvex GA-Planes formulations come with guarantees that all local optima are also global (Sahiner et al., 2024).

**Relation to Prior Models.** Without any convexity restrictions, the GA-Planes family includes many previously proposed models as special cases:

- NeRF (Mildenhall et al., 2020): D

- Plenoxels (Sara Fridovich-Keil and Alex Yu et al., 2022), DVGO (Sun et al., 2022): $D(\mathbf{e}_{123})$

- TensoRF (Chen et al., 2022): $D((\mathbf{e}_1 \circ \mathbf{e}_{23}) \odot (\mathbf{e}_2 \circ \mathbf{e}_{13}) \odot (\mathbf{e}_3 \circ \mathbf{e}_{12}))$

- Tri-Planes (Chan et al., 2022): $D(\mathbf{e}_{12} + \mathbf{e}_{13} + \mathbf{e}_{23})$

- K-Planes (Fridovich-Keil et al., 2023): $D(\mathbf{e}_{12} \circ \mathbf{e}_{13} \circ \mathbf{e}_{23})$

- MERF (Reiser et al., 2023): $D(\mathbf{e}_{12} + \mathbf{e}_{13} + \mathbf{e}_{23} + \mathbf{e}_{123})$

Of these, all except for TensoRF and K-Planes are compatible with convex optimization towards any convex objective. Note that different models may use different decoder architectures for D, including both linear and MLP decoders and additional decoder inputs such as encoded viewing direction and/or positionally-encoded coordinates.

**Convex Neural Networks.** Given a data matrix $X \in \mathbb{R}^{n \times d}$ and labels $y \in \mathbb{R}^n$, a 2-layer nonconvex ReLU MLP approximates $y$ as

$$y \approx \sum_{j=1}^{m} (XU_j)_+ \alpha_j, \tag{12}$$

where $m$ is the number of hidden neurons and $U$ and $\alpha$ are the first and second linear layer weights, respectively. (Pilanci & Ergen, 2020) proposed to instead approximate $y$ as

$$y \approx \sum_{i=1}^{P} D_i X(v_i - w_i), \tag{13}$$

subject to $(2D_i - I_n)Xv_i \geq 0$ and $(2D_i - I_n)Xw_i \geq 0$ for all $i$. The parameters $v$ and $w$ in Equation (13) replace the first and second layer weights $U$ and $\alpha$ from the nonconvex formulation in Equation (12) (optimal values of $U$ and $\alpha$ can be recovered from optimal values of $v$ and $w$). Here $D_i$ represent different possible activation patterns of the hidden neurons as $\{D_i\}_{i=1}^{P} := \{\text{Diag}(\mathbb{1}[Xu \geq 0]) : u \in \mathbb{R}^d\}$, which is the finite set of hyperplane arrangement patterns obtained for all possible $u \in \mathbb{R}^d$. We can sample different $u$'s to find all distinct activation patterns $\{D_i\}_{i=1}^{P}$, where $P$ is the number of regions in the partitioned input space. Enumerating all such patterns would yield an exact equivalence with the global minimizer of the nonconvex ReLU MLP in Equation (12), but may be complicated or intractable due to memory limitations. Subsampling $\tilde{P}$ patterns results in a convex program with tractable size, whose solution is one of the stationary points of the original non-convex problem (Pilanci & Ergen, 2020). We apply this idea to create convex and semiconvex GA-Planes models by convexifying the feature decoder MLP according to this procedure.

### A.2. Proof of Theorems

**A general note on proofs.** In order to represent a matrix $M \in \mathbb{R}^{m \times n}$ with an implicit model, we compute $D(f(q))$ for $q = (k, l)$, $\forall k \in \{1, \ldots, m\}$, $\forall l \in \{1, \ldots, n\}$. Considering line feature grids with resolutions matching $m, n$; the features will become $\mathbf{e}_1 = (\mathbf{g}_1)_k$, $\mathbf{e}_2 = (\mathbf{g}_2)_l$ for $q = (k, l)$ otherwise they will be $\mathbf{e}_1 = \varphi(\mathbf{g}_1)_k$, $\mathbf{e}_2 = \varphi(\mathbf{g}_2)_l$ where $\varphi(\mathbf{g}_1), \varphi(\mathbf{g}_2)$ now have resolutions $m, n$ after interpolation through $\varphi$. Here $\varphi$ can be any interpolation scheme with linear weighting of inputs, e.g. nearest neighbor, (bi)linear, (bi)cubic, spline, Gaussian, sinc, etc. For the simplicity of notation, we omit $\varphi$ in line feature grids, and only apply it to the plane feature grid $\mathbf{g}_{12}$, which has lower resolution by design. The

proofs consider $\mathbf{g}_1, \mathbf{g}_2 \in \mathbb{R}^{r_1 \times d_1}$ and $\mathbf{g}_{12} \in \mathbb{R}^{r_2 \times r_2 \times d_1}$ (equal feature dimensions, different resolutions), resulting in the matrix representation $\hat{M} \in \mathbb{R}^{r_1 \times r_1}$ (the case where $m = n = r_1$). Note that if the interpolation is done by a method other than nearest neighbor, this may allow a (convex or nonconvex) MLP decoder to increase the rank beyond $r_1$. We derive expressions for $\hat{M}$ implied by different GA-Planes variations in the parts that follow. The coordinate-wise optimization objective (in the case of a directly supervised mean-square-error loss) corresponds to minimizing the Frobenius norm of the ground truth matrix $M$ and its approximation $\hat{M}$.

### A.2.1. PROOF OF THEOREM 1

The forward mapping of the model $D(\mathbf{e}_1 + \mathbf{e}_2)$ is:

$$\tilde{y}(q) = D(\mathbf{e}_1 + \mathbf{e}_2) = \alpha^\top (\mathbf{e}_1 + \mathbf{e}_2) = \alpha^\top ((\mathbf{g}_1)_k + (\mathbf{g}_2)_l) = \sum_{i=1}^{d_1} \alpha_i ((\mathbf{g}_1)_{k,i} + (\mathbf{g}_2)_{l,i}), \tag{14}$$

where $(\mathbf{g}_1)_k, (\mathbf{g}_1)_l \in \mathbb{R}^{d_1 \times 1}$.

In matrix form,

$$\hat{M} = \sum_{i=1}^{d_1} \alpha_i ((\mathbf{g}_1)_i \mathbb{1}^\top + \mathbb{1}(\mathbf{g}_2)_i^\top) = \sum_{i=1}^{d_1} \alpha_i (\mathbf{g}_1)_i \mathbb{1}^\top + \sum_{i=1}^{d_1} \alpha_i \mathbb{1}(\mathbf{g}_2)_i^\top. \tag{15}$$

Defining $U := \mathbf{g}_1 \mathrm{diag}(\alpha), U \in \mathbb{R}^{r_1 \times d_1}$ and $V := \mathbf{g}_2 \mathrm{diag}(\alpha), V \in \mathbb{R}^{r_1 \times d_1}$, this can be expressed as

$$\hat{M} = U \mathbb{1}_{r_1 \times d_1}^\top + \mathbb{1}_{r_1 \times d_1} V^\top. \tag{16}$$

Note that the resulting matrix $\hat{M} \in \mathbb{R}^{r_1 \times r_1}$ has rank at most 2—very limited expressivity—regardless of the resolution $r_1$. This is because the all-ones matrix is rank 1, and a product of matrices cannot have higher rank than either of its factors.

Similarly for the multiplicative representation $D(\mathbf{e}_1 \circ \mathbf{e}_2)$, the mapping is

$$\tilde{y}(q) = D(\mathbf{e}_1 \circ \mathbf{e}_2) = \alpha^\top (\mathbf{e}_1 \circ \mathbf{e}_2) = \alpha^\top ((\mathbf{g}_1)_k \circ (\mathbf{g}_2)_l) = \sum_{i=1}^{d_1} \alpha_i (\mathbf{g}_1)_{k,i} (\mathbf{g}_2)_{l,i}, \tag{17}$$

where $(\mathbf{g}_1)_k, (\mathbf{g}_1)_l \in \mathbb{R}^{d_1 \times 1}$. In matrix form,

$$\hat{M} = \sum_{i=1}^{d_1} \alpha_i (\mathbf{g}_1)_i (\mathbf{g}_2)_i^\top = \mathbf{g}_1 \mathrm{diag}(\alpha) \mathbf{g}_2^\top. \tag{18}$$

Defining $U := \mathbf{g}_1 \mathrm{diag}(\alpha), U \in \mathbb{R}^{r_1 \times d_1}$ and $V := \mathbf{g}_2, V \in \mathbb{R}^{r_1 \times d_1}$, this can be expressed as

$$\hat{M} = U V^\top, \tag{19}$$

which is the optimal rank-$d_1$ decomposition.

### A.2.2. PROOF OF THEOREM 2

The forward mapping of the model $D(\mathbf{e}_1 + \mathbf{e}_2 + \mathbf{e}_{12})$ becomes:

$$\tilde{y}(q) = D(\mathbf{e}_1 + \mathbf{e}_2 + \mathbf{e}_{12}) = \alpha^\top (\mathbf{e}_1 + \mathbf{e}_2 + \mathbf{e}_{12}) = \alpha^\top ((\mathbf{g}_1)_k + (\mathbf{g}_2)_l + \varphi(\mathbf{g}_{12})_{k,l}) \tag{20}$$

$$= \sum_{i=1}^{d_1} \alpha_i ((\mathbf{g}_1)_{k,i} + (\mathbf{g}_2)_{l,i}) + \sum_{i=1}^{d_1} \alpha_i \varphi(\mathbf{g}_{12})_{k,l,i}, \tag{21}$$

where $(\mathbf{g}_1)_k, (\mathbf{g}_1)_l, \varphi(\mathbf{g}_{12})_{k,l} \in \mathbb{R}^{d_1 \times 1}$. In matrix form,

$$\hat{M} = \sum_{i=1}^{d_1} \alpha_i ((\mathbf{g}_1)_i \mathbb{1}^\top + \mathbb{1}(\mathbf{g}_2)_i^\top) + \sum_{i=1}^{d_1} \alpha_i \varphi(\mathbf{g}_{12})_i. \tag{22}$$

Noting that the first term is the same as in Equation (15) and defining $L := \mathbf{g}_{12}\alpha$, $L \in \mathbb{R}^{r_2 \times r_2}$, we reach the expression

$$\hat{M} = U\mathbb{1}_{d_1 \times r_1} + \mathbb{1}_{r_1 \times d_1} V^\top + \varphi(L), \tag{23}$$

since $\sum_{i=1}^{d_1} \alpha_i \varphi(\mathbf{g}_{12})_i = \varphi(\sum_{i=1}^{d_1} \alpha_i(\mathbf{g}_{12})_i) = \varphi(\mathbf{g}_{12}\alpha)$, following the linearity of the upsampling function $\varphi$. Note that in the definition of $L$ there is a tensor-vector product that effectively takes a dot product along the last (feature) dimension.

Similarly for the multiplicative representation $\mathrm{D}(\mathbf{e}_1 \circ \mathbf{e}_2 + \mathbf{e}_{12})$, the mapping is

$$\tilde{y}(q) = \mathrm{D}(\mathbf{e}_1 \circ \mathbf{e}_2 + \mathbf{e}_{12}) = \alpha^\top(\mathbf{e}_1 \circ \mathbf{e}_2 + \mathbf{e}_{12}) \tag{24}$$

$$= \alpha^\top((\mathbf{g}_1)_k \circ (\mathbf{g}_2)_l + \varphi(\mathbf{g}_{12})_{k,l}) = \sum_{i=1}^{d_1} \alpha_i(\mathbf{g}_1)_{k,i}(\mathbf{g}_2)_{l,i} + \sum_{i=1}^{d_1} \alpha_i \varphi(\mathbf{g}_{12})_{k,l,i}. \tag{25}$$

In matrix notation, we have

$$\hat{M} = \sum_{i=1}^{d_1} \alpha_i((\mathbf{g}_1)_i(\mathbf{g}_2)_i^\top) + \sum_{i=1}^{d_1} \alpha_i \varphi(\mathbf{g}_{12})_i. \tag{26}$$

Following Equation (18) and using the same definition of $L$, the final expression becomes

$$\hat{M} = UV^\top + \varphi(L). \tag{27}$$

### A.2.3. PROOF OF THEOREM 3

For a 2-layer convex MLP with hidden size $h$, denote the trainable first layer weights as $W \in \mathbb{R}^{h \times d_1}$ and the gating weights as $\overline{W} \in \mathbb{R}^{h \times d_1}$ (which are fixed at random initialization). We will handle three different cases for merging the interpolated features: multiplication ($\circ$), addition ($+$), and concatenation ($\odot$).

The forward mapping of the multiplicative model using a convex MLP, $\mathrm{D}(\mathbf{e}_1 \circ \mathbf{e}_2)$ at $q = (k, l)$ is

$$\tilde{y}(q) = \mathbb{1}_h^\top \left((W((\mathbf{g}_1)_k \circ (\mathbf{g}_2)_l)) \circ \mathbb{1}\left[\overline{W}((\mathbf{g}_1)_k \circ (\mathbf{g}_2)_l) \geq 0\right]\right) \tag{28}$$

$$= \sum_{i=1}^{h} \left(\sum_{j=1}^{d_1} W_{i,j}(\mathbf{g}_1)_{k,j}(\mathbf{g}_2)_{l,j}\right) \mathbb{1}\left[\sum_{j=1}^{d_1} \overline{W}_{i,j}(\mathbf{g}_1)_{k,j}(\mathbf{g}_2)_{l,j} \geq 0\right], \tag{29}$$

where $\circ$ denotes elementwise multiplication (Hadamard product). The resulting matrix decomposition can then be written as

$$\hat{M} = \sum_{i=1}^{h} \left(\sum_{j=1}^{d_1} W_{i,j}(\mathbf{g}_1)_j(\mathbf{g}_2)_j^\top\right) \circ \mathbb{1}\left[\sum_{j=1}^{d_1} \overline{W}_{i,j}(\mathbf{g}_1)_j(\mathbf{g}_2)_j^\top \geq 0\right]. \tag{30}$$

Now, we define the masking matrix $B_i = \mathbb{1}\left[\sum_{j=1}^{d_1} \overline{W}_{i,j}(\mathbf{g}_1)_j(\mathbf{g}_2)_j^\top \geq 0\right]$ and the eigenvectors $U_j = (\mathbf{g}_1)_j$, $V_j = (\mathbf{g}_2)_j$ to reach the expression from the theorem statement:

$$\hat{M} = \sum_{i,j} W_{i,j} U_j V_j^\top \circ B_i. \tag{31}$$

When the model uses additive features as in $\mathrm{D}(\mathbf{e}_1 + \mathbf{e}_2)$, and D is a convex MLP, the prediction is

$$\tilde{y}(q) = \mathbb{1}_h^\top \left((W((\mathbf{g}_1)_k + (\mathbf{g}_2)_l)) \circ \mathbb{1}\left[\overline{W}((\mathbf{g}_1)_k + (\mathbf{g}_2)_l) \geq 0\right]\right) \tag{32}$$

$$= \sum_{i=1}^{h} \left(\sum_{j=1}^{d_1} W_{i,j}((\mathbf{g}_1)_{k,j} + (\mathbf{g}_2)_{l,j})\right) \mathbb{1}\left[\sum_{j=1}^{d_1} \overline{W}_{i,j}((\mathbf{g}_1)_{k,j} + (\mathbf{g}_2)_{l,j}) \geq 0\right]. \tag{33}$$

The resulting matrix decomposition can then be written as

$$\hat{M} = \sum_{i=1}^{h} \left( \sum_{j=1}^{d_1} W_{i,j}((\mathbf{g}_1)_j \mathbb{1}_{r_1}^\top + \mathbb{1}_{r_1}(\mathbf{g}_2)_j^\top) \right) \mathbb{1} \left[ \sum_{j=1}^{d_1} \overline{W}_{i,j}((\mathbf{g}_1)_j \mathbb{1}_{r_1}^\top + \mathbb{1}_{r_1}(\mathbf{g}_2)_j^\top) \geq 0 \right]. \tag{34}$$

Defining $B_i = \mathbb{1}\left[ \sum_{j=1}^{d_1} \overline{W}_{i,j}((\mathbf{g}_1)_j \mathbb{1}_{r_1}^\top + \mathbb{1}_{r_1}(\mathbf{g}_2)_j^\top) \geq 0 \right]$, $U_j = (\mathbf{g}_1)_j$, $V_j = (\mathbf{g}_2)_j$, we reach the final expression:

$$\hat{M} = \sum_{i,j} W_{i,j} \left( U_j \mathbb{1}_{r_1}^\top + \mathbb{1}_{r_1} V_j^\top \right) \circ B_i. \tag{35}$$

Finally, we show that concatenation of features results in a very similar expression to Equation (35).

When the model uses concatenated features as in $\mathrm{D}(\mathbf{e}_1 \odot \mathbf{e}_2)$, and D is a convex MLP (with trainable weights $W \in \mathbb{R}^{h \times 2d_1}$ and fixed gates $\overline{W} \in \mathbb{R}^{h \times 2d_1}$), the prediction at a point $q$ is

$$\tilde{y}(q) = \mathbb{1}_h^\top \left( (W((\mathbf{g}_1)_k \odot (\mathbf{g}_2)_l)) \circ \mathbb{1}\left[ \overline{W}((\mathbf{g}_1)_k \odot (\mathbf{g}_2)_l) \geq 0 \right] \right). \tag{36}$$

Denoting the weights and gates each as a concatenation of 2 matrices, $W = (W_1 \odot W_2)$, $\overline{W} = (\overline{W}_1 \odot \overline{W}_2)$, where $W_1, W_2, \overline{W}_1, \overline{W}_2 \in \mathbb{R}^{h \times d_1}$, we have the following expression:

$$\tilde{y}(q) = \sum_{i=1}^{h} \left( \sum_{j=1}^{d_1} W_{1i,j}(\mathbf{g}_1)_{k,j} + W_{2i,j}(\mathbf{g}_2)_{l,j} \right) \mathbb{1} \left[ \sum_{j=1}^{d_1} \overline{W}_{1i,j}(\mathbf{g}_1)_{k,j} + \overline{W}_{2i,j}(\mathbf{g}_2)_{l,j} \geq 0 \right]. \tag{37}$$

Following similar steps as for the additive case, we express the matrix decomposition as

$$\hat{M} = \sum_{i,j} (W_{1i,j} U_j \mathbb{1}_{r_1}^\top + W_{2i,j} \mathbb{1}_{r_1} V_j^\top) \circ B_i, \tag{38}$$

where $B_i = \mathbb{1}\left[ \sum_{j=1}^{d_1} \overline{W}_{1i,j}(\mathbf{g}_1)_j \mathbb{1}_{r_1}^\top + \overline{W}_{2i,j} \mathbb{1}_{r_1}(\mathbf{g}_2)_j^\top \geq 0 \right]$, $U_j = (\mathbf{g}_1)_j$, $V_j = (\mathbf{g}_2)_j$.

In all these representations, a low-rank matrix is multiplied elementwise with a binary mask $B_i$, which makes the maximum attainable rank $r_1$. Thus, with a convex MLP decoder, rank of $\hat{M}$ is limited by the resolution of the feature grids.

### A.2.4. PROOF OF THEOREM 4

For a standard 2-layer ReLU MLP with hidden size $h$, denote the trainable first and second layer weights as $W \in \mathbb{R}^{h \times d_1}$, $\alpha \in \mathbb{R}^{h \times 1}$. We will handle three different cases for merging the interpolated features: multiplication ($\circ$), addition ($+$), and concatenation ($\odot$).

The forward mapping of the multiplicative model using a standard nonconvex MLP, $\mathrm{D}(\mathbf{e}_1 \circ \mathbf{e}_2)$ is:

$$\tilde{y}(q) = \alpha^\top [W((\mathbf{g}_1)_k \circ (\mathbf{g}_2)_l)]_+ \tag{39}$$

$$= \sum_{i=1}^{h} \alpha_i \left[ \sum_{j=1}^{d_1} W_{i,j}(\mathbf{g}_1)_{k,j}(\mathbf{g}_2)_{l,j} \right]_+. \tag{40}$$

The resulting matrix decomposition can then be written as

$$\hat{M} = \sum_{i=1}^{h} \alpha_i \left( \sum_{j=1}^{d_1} W_{i,j}(\mathbf{g}_1)_j(\mathbf{g}_2)_j^\top \right)_+ = \sum_{i=1}^{h} \alpha_i \left( \sum_{j=1}^{d_1} W_{i,j} U_j V_j^\top \right)_+, \tag{41}$$

with $U_j = (\mathbf{g}_1)_j$, $V_j = (\mathbf{g}_2)_j$.

When the model uses additive features as in $D(\mathbf{e}_1 + \mathbf{e}_2)$, the prediction is

$$\tilde{y}(q) = \alpha^\top \left[ W((\mathbf{g}_1)_k + (\mathbf{g}_2)_l) \right]_+ \tag{42}$$

$$= \sum_{i=1}^h \alpha_i \left[ \sum_{j=1}^{d_1} W_{i,j}((\mathbf{g}_1)_{k,j} + (\mathbf{g}_2)_{l,j}) \right]_+ . \tag{43}$$

The resulting matrix decomposition can then be written as

$$\hat{M} = \sum_{i=1}^h \alpha_i \left( \sum_{j=1}^{d_1} W_{i,j}((\mathbf{g}_1)_j \mathbb{1}_{r_1}^\top + \mathbb{1}_{r_1}(\mathbf{g}_2)_j^\top) \right)_+ = \sum_{i=1}^h \alpha_i \left( \sum_{j=1}^{d_1} W_{i,j}(U_j \mathbb{1}_{r_1}^\top + \mathbb{1}_{r_1} V_j^\top) \right)_+ , \tag{44}$$

again with $U_j = (\mathbf{g}_1)_j$, $V_j = (\mathbf{g}_2)_j$.

Finally, we show that concatenation of features results in a very similar expression.

When the model uses concatenated features as in $D(\mathbf{e}_1 \odot \mathbf{e}_2)$ and D is a standard nonconvex MLP (with trainable weights $W \in \mathbb{R}^{h \times 2d_1}$ and $\alpha \in \mathbb{R}^{h \times 1}$), the prediction at a point $q$ is

$$\tilde{y}(q) = \alpha^\top \left[ (W((\mathbf{g}_1)_k \odot (\mathbf{g}_2)_l))_+ \right] . \tag{45}$$

Denoting the hidden layer weights as a concatenation of 2 matrices, $W = (W_1 \odot W_2)$, where $W_1, W_2 \in \mathbb{R}^{h \times d_1}$, we have the following expression:

$$\tilde{y}(q) = \sum_{i=1}^h \alpha_i \left( \sum_{j=1}^{d_1} W_{1i,j}(\mathbf{g}_1)_{k,j} + W_{2i,j}(\mathbf{g}_2)_{l,j} \right)_+ . \tag{46}$$

Following similar steps, we express the matrix decomposition as

$$\hat{M} = \sum_{i=1}^h \alpha_i \left( \sum_{j=1}^{d_1} W_{1i,j}(\mathbf{g}_1)_j \mathbb{1}_{r_1}^\top + W_{2i,j} \mathbb{1}_{r_1}(\mathbf{g}_2)_j^\top \right)_+ \tag{47}$$

$$= \sum_{i=1}^h \alpha_i \left( \sum_{j=1}^{d_1} W_{1i,j} U_j \mathbb{1}_{r_1}^\top + W_{2i,j} \mathbb{1}_{r_1} V_j^\top \right)_+ , \tag{48}$$

where $U_j = (\mathbf{g}_1)_j$, $V_j = (\mathbf{g}_2)_j$.

By a similar argument to Appendix A.2.3, the maximum attainable rank of all three representations derived here is limited by $r_1$.

### A.3. Interpolation Comparison

We present 2D image fitting experiments with the *astronaut* image from SciPy, validating matrix completion analysis summarized in Table 1. We compare 2D GA-Planes models of the form $D(\mathbf{e}_1 \circ \mathbf{e}_2)$ (solid colorful lines) and $D(\mathbf{e}_1 + \mathbf{e}_2)$ (dotted colorful lines) with the optimal low-rank approximation provided by singular value decomposition (solid black line) in Figure 7. The same experiment is repeated for linear interpolation into the vector (line) features (left) versus nearest neighbor interpolation (right, same as theorems). In this experiment we find qualitatively similar results regardless of the type of interpolation, with slightly better performance using linear interpolation; in our 3D experiments we use (bi/tri)linear interpolation.

As expected, we find that a linear decoder model with multiplication dramatically outperforms its additive counterpart, which does not improve with increasing model size. We also find that 2D GA-Planes models with MLP decoders can match or exceed the compression performance of the optimal low-rank representation found by singular value decomposition (SVD), especially when using a nonconvex MLP. This is a testament to the capacity of an MLP decoder to increase representation rank using fewer parameters than a traditional low-rank decomposition, as well as to the resolution compressibility of natural images.

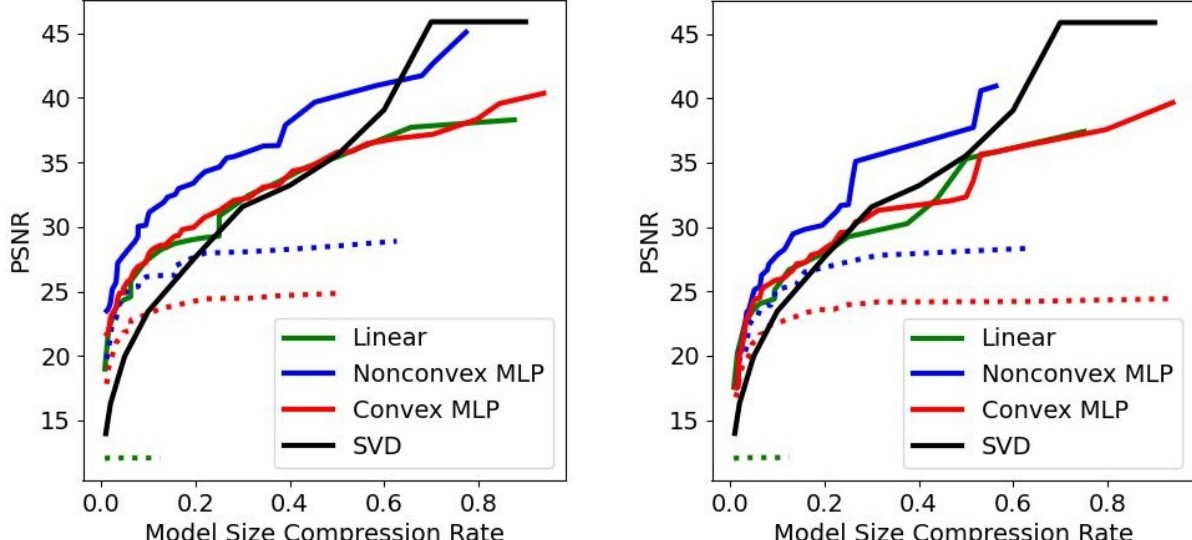

**Figure 7:** 2D image fitting experiments matching the setting of our theoretical results, with a GA-Planes version using only vector (line) features and a decoder as specified in the legend. Solid lines denote features combined by multiplication; dotted lines use addition. Left: linear interpolation of features; Right: nearest neighbor interpolation of features. For the SVD baseline we use low-rank factors whose resolution matches the target image, so no interpolation is needed (this, and the use of a nonlinear decoder, is why in some cases 2D GA-Planes can outperform SVD).

### A.4. Lower Bounds

Based on the matrix completion theorems and their summary in Table 1, we present lower bounds on the Frobenius norm errors of each 2D GA-Planes model. We denote the optimal fitting error of the linear and MLP decoder models by $E_{linear}(\mathrm{D}(f(q)))$ and $E_{MLP}(\mathrm{D}(f(q)))$ for different feature combinations $f(q)$. For models with a linear decoder,

$$E_{linear}(\mathrm{D}(\mathbf{e}_1 + \mathbf{e}_2)) \geq \sigma_2(M) \tag{49}$$

$$E_{linear}(\mathrm{D}(\mathbf{e}_1 \circ \mathbf{e}_2)) \geq \sigma_k(M) \tag{50}$$

$$E_{linear}(\mathrm{D}(\mathbf{e}_1 \circ \mathbf{e}_2 + \mathbf{e}_{12})) \geq \sigma_k(M - \varphi(L^*)), \tag{51}$$

where $L^*$ is a downsampled version of the target $M$, at the same resolution as the feature grid $\mathbf{g}_{12}$.

For models with convex or nonconvex MLP decoders,

$$E_{MLP}(\mathrm{D}(\mathbf{e}_1 + \mathbf{e}_2)) \geq \sigma_{r_1}(M) \tag{52}$$

$$E_{MLP}(\mathrm{D}(\mathbf{e}_1 \circ \mathbf{e}_2)) \geq \sigma_{r_1}(M) \tag{53}$$

$$E_{MLP}(\mathrm{D}(\mathbf{e}_1 \circ \mathbf{e}_2 + \mathbf{e}_{12})) \geq \sigma_{r_1}(M - \varphi(L^*)). \tag{54}$$

We can see from these bounds that the approximation error of a model can be reduced dramatically by the introduction of a convex or nonconvex MLP decoder, depending on the singular value decay of the target image $M$ or its high-frequency residual.

### A.5. Results for all Nerfstudio-Blender Scenes

In this section, we provide per-scene pareto-optimal curves and qualitative rendering comparisons on various scenes from the Blender dataset. We highlight the superior performance of GA-Planes with limited number of parameters by comparing the smallest K-Planes, TensoRF and GA-Planes models.

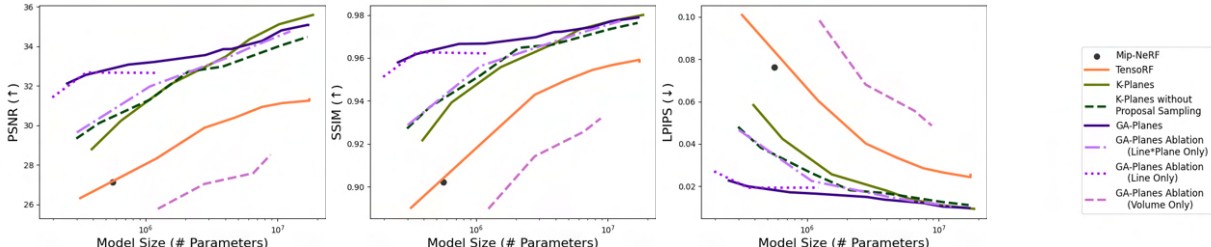

**Figure 8:** Results on radiance field reconstruction. Nonconvex GA-Planes (with feature multiplication) offers the most efficient representation: when the model is large it performs comparably to the state of the art models, but when model size is reduced it retains higher performance than other models. Here all models are trained for the same number of epochs on the *lego* scene.

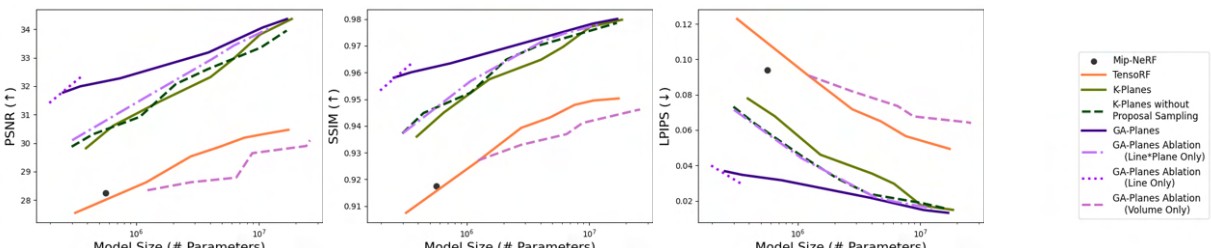

**Figure 9:** Results on radiance field reconstruction. Nonconvex GA-Planes (with feature multiplication) offers the most efficient representation: when the model is large it performs comparably to the state of the art models, but when model size is reduced it retains higher performance than other models. Here all models are trained for the same number of epochs on the *chair* scene.

## A.6. Model configurations used for experiments

### A.6.1. RADIANCE FIELD MODELING

In our NeRFStudio experiments, we find that because GA-Planes contains features with different dimensionalities (line, plane, and volume), it experiences little loss in quality over a wide range of model sizes. For small model sizes, we allocate most of the model memory to the line features, since their spatial resolution grows linearly with parameter count. As the parameter budget grows, we allocate more parameters to the plane features, whereas the performance of the line-only model stagnates with increasing size. Similarly, as model size grows even further we allocate more parameters to the volume features, whose memory footprint grows cubically with spatial resolution.

The original K-planes model uses 2 proposal networks with different resolutions (as noted in Table 5) and a fixed channel dimension of 8 for both. The resolutions and channel dimensions for either K-planes model (with vs. without proposal sampling) refer to $r_2$ and $d_2$, respectively. TensoRF model resolutions and channel dimensions can be interpreted in a similar way, since their feature combination dictates that $d_1 = d_2$ and they initialize the line and plane grids with the same resolution. The only nuance is that TensoRF constructs separate features for color and density decoding. Hence, the channel dimensions for density and color features are listed. Instead of a multiresolution scheme, TensoRF starts from the base resolution of $r_1 = r_2 = 128$ and upsamples the grids to reach the final resolutions on Table 5. Resolutions listed under GA-Planes should be interpreted as $[r_1, r_2, r_3]$; channel dimensions as $[d_1, d_2, d_3]$. For all models that use the multiresolution scheme, the base resolutions (i.e. $[r_1, r_2, r_3]$) are multiplied with the upsampling factors. For instance, a base resolution $[r_1, r_2, r_3]$ with channel dimensions $[d_1, d_2, d_3]$ and multiresolution copies $[m_1, m_2, m_3]$ will generate the grids of GA-Planes as follows: Linear feature grids $\mathbf{g}_1, \mathbf{g}_2, \mathbf{g}_3$ will have the shapes $\{[m_1 r_1, d_1], [m_2 r_1, d_1], [m_3 r_1, d_1]\}$, plane grids $\mathbf{g}_{12}, \mathbf{g}_{23}, \mathbf{g}_{13}$ will have the shapes $\{[m_1 r_2, m_1 r_2, d_2], [m_2 r_2, m_2 r_2, d_2], [m_3 r_2, m_3 r_2, d_2]\}$, and the volume grid $\mathbf{g}_{123}$ will have the shape $[r_3, r_3, r_3, d_3]$. Although we don't use multiresolution copies for the volume grid in GA-Planes, we do use multiresolution for the volume-only GA-Planes ablation. The resolution for that model refers to $r_3$, and the channel dimensions are also allowed to vary for each resolution (unlike other variants with multiresolution, where the feature dimension is fixed across resolutions). If we denote these varying feature dimensions as $[d_{3a}, d_{3b}, d_{3c}]$, the multiresolution copies of the volume grids

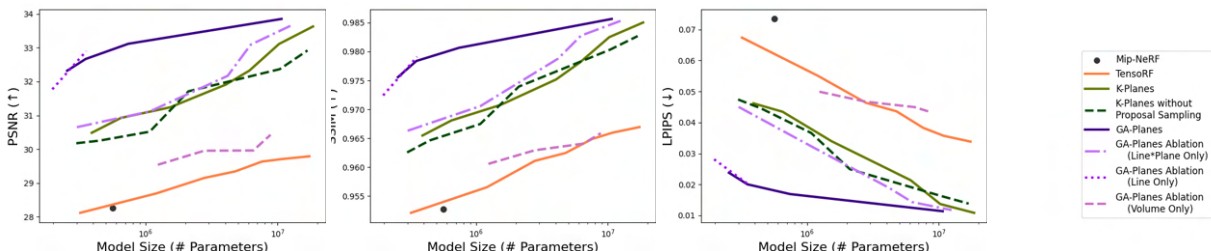

**Figure 10:** Results on radiance field reconstruction. Nonconvex GA-Planes (with feature multiplication) offers the most efficient representation: when the model is large it performs comparably to the state of the art models, but when model size is reduced it retains higher performance than other models. Here all models are trained for the same number of epochs on the *mic* scene.

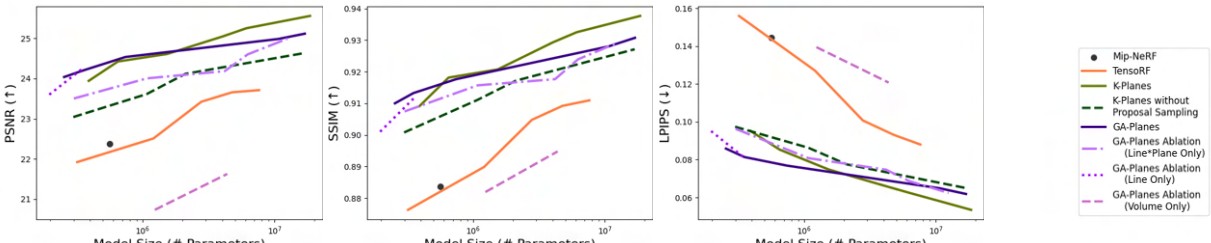

**Figure 11:** Results on radiance field reconstruction. Nonconvex GA-Planes (with feature multiplication) offers the most efficient representation: when the model is large it performs comparably to the state of the art models, but when model size is reduced it retains higher performance than other models. Here all models are trained for the same number of epochs on the *drums* scene.

will have the shapes $\{[m_1r_3, m_1r_3, m_1r_3, d_{3a}], [m_2r_3, m_2r_3, m_2r_3, d_{3b}], [m_3r_3, m_3r_3, m_3r_3, d_{3c}]\}$.

### A.6.2. 3D SEGMENTATION

GA-Planes model uses feature dimensions $[d_1, d_2, d_3] = [36, 24, 8]$ (with $\odot$) or $[d_1, d_2, d_3] = [25, 25, 8]$ (with $\circ$) and resolutions $[r_1, r_2, r_3] = [128, 32, 24]$. Multiresolution grids are not used for this task since density prediction can be achieved by a simpler architecture. The model size is 0.22 M. Tri-Planes model has the feature dimension $d_2 = 4$, and resolution $r_2 = 128$ resulting in a total number of parameters of 0.2 M. Note that we fix these sizes across (non/semi)convex formulations, which causes slight variations in the size of the decoder, however, the grids constitute the most number of parameters, making this effect negligible.

### A.6.3. VIDEO SEGMENTATION

GA-Planes model uses feature dimensions $[d_1, d_2, d_3] = [32, 16, 8]$ and resolutions $[r_1, r_2, r_3] = [128, 128, 64]$. When the features are combined by multiplication in the nonconvex model, $d_1 = d_2 = 16$. Multiresolution grids are not used for this task. The model size is 2.9 M. Tri-Planes model has the feature dimension $d_2 = 59$, and resolution $r_2 = 128$ resulting in a total number of parameters of 2.9 M.

### A.7. Memory and Compute Times

We plot the training times, memory sizes and PSNR metrics of GA-Planes and K-Planes for radiance field reconstruction task in Figure 20, using the same configurations as in Figure 3. Our method does not incur a memory or a compute overhead, however the training times vary due to the choice of feature dimensions. GA-Planes and K-Planes achieve the same training time with the same feature dimension, and both models can be configured to achieve faster training by reducing feature dimensions.

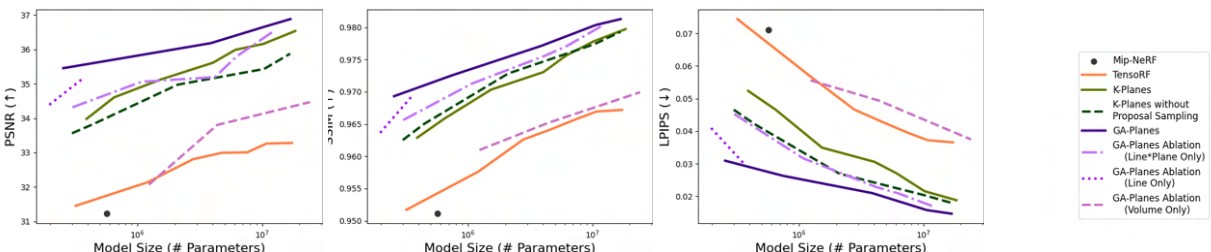

**Figure 12:** Results on radiance field reconstruction. Nonconvex GA-Planes (with feature multiplication) offers the most efficient representation: when the model is large it performs comparably to the state of the art models, but when model size is reduced it retains higher performance than other models. Here all models are trained for the same number of epochs on the *hotdog* scene.

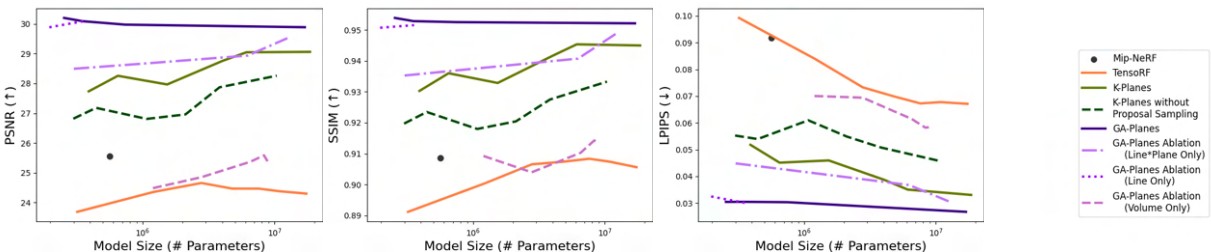

**Figure 13:** Results on radiance field reconstruction. Nonconvex GA-Planes (with feature multiplication) offers the most efficient representation: when the model is large it performs comparably to the state of the art models, but when model size is reduced it retains higher performance than other models. Here all models are trained for the same number of epochs on the *materials* scene.

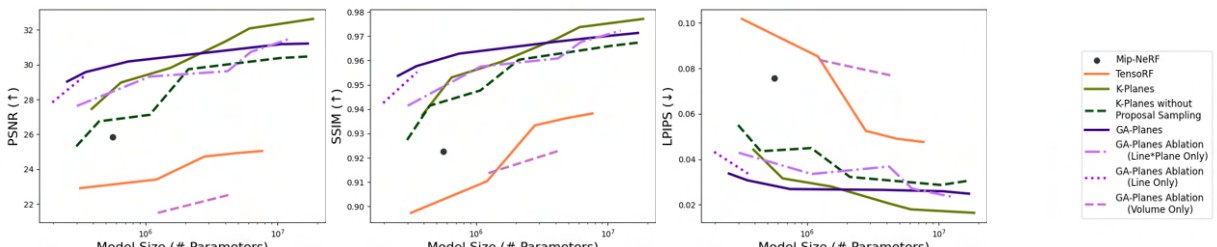

**Figure 14:** Results on radiance field reconstruction. Nonconvex GA-Planes (with feature multiplication) offers the most efficient representation: when the model is large it performs comparably to the state of the art models, but when model size is reduced it retains higher performance than other models. Here all models are trained for the same number of epochs on the *ficus* scene.

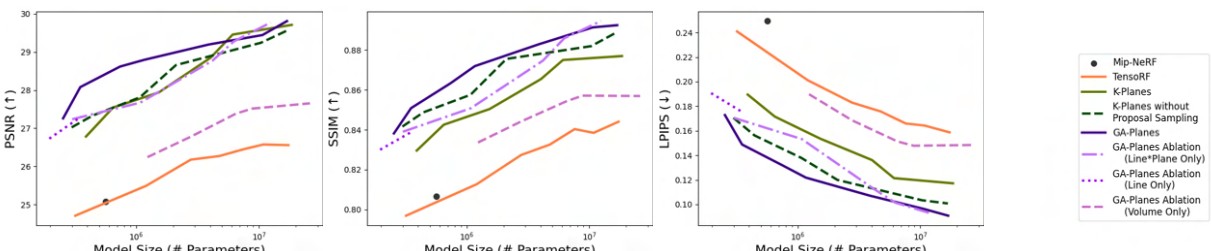

**Figure 15:** Results on radiance field reconstruction. Nonconvex GA-Planes (with feature multiplication) offers the most efficient representation: when the model is large it performs comparably to the state of the art models, but when model size is reduced it retains higher performance than other models. Here all models are trained for the same number of epochs on the *ship* scene.

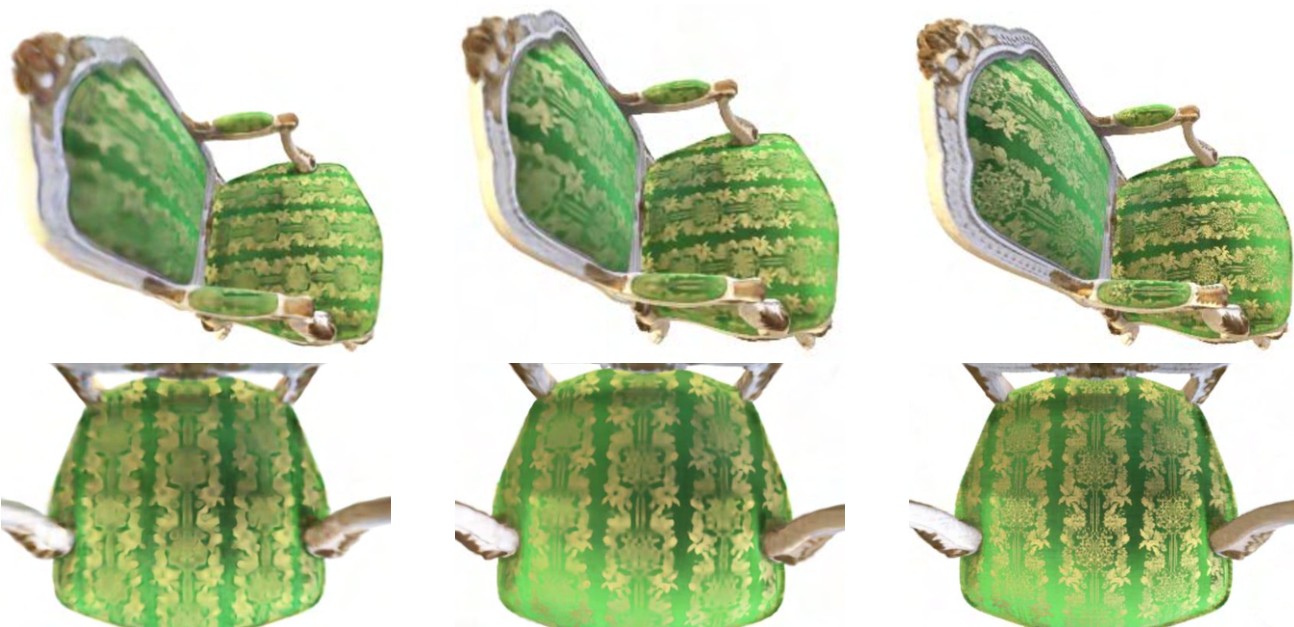

**Figure 16:** Rendering comparison for the *chair* scene: TensoRF on the left (0.32 M parameters), K-Planes in the middle (0.39 M parameters), GA-Planes on the right (0.25 M parameters).

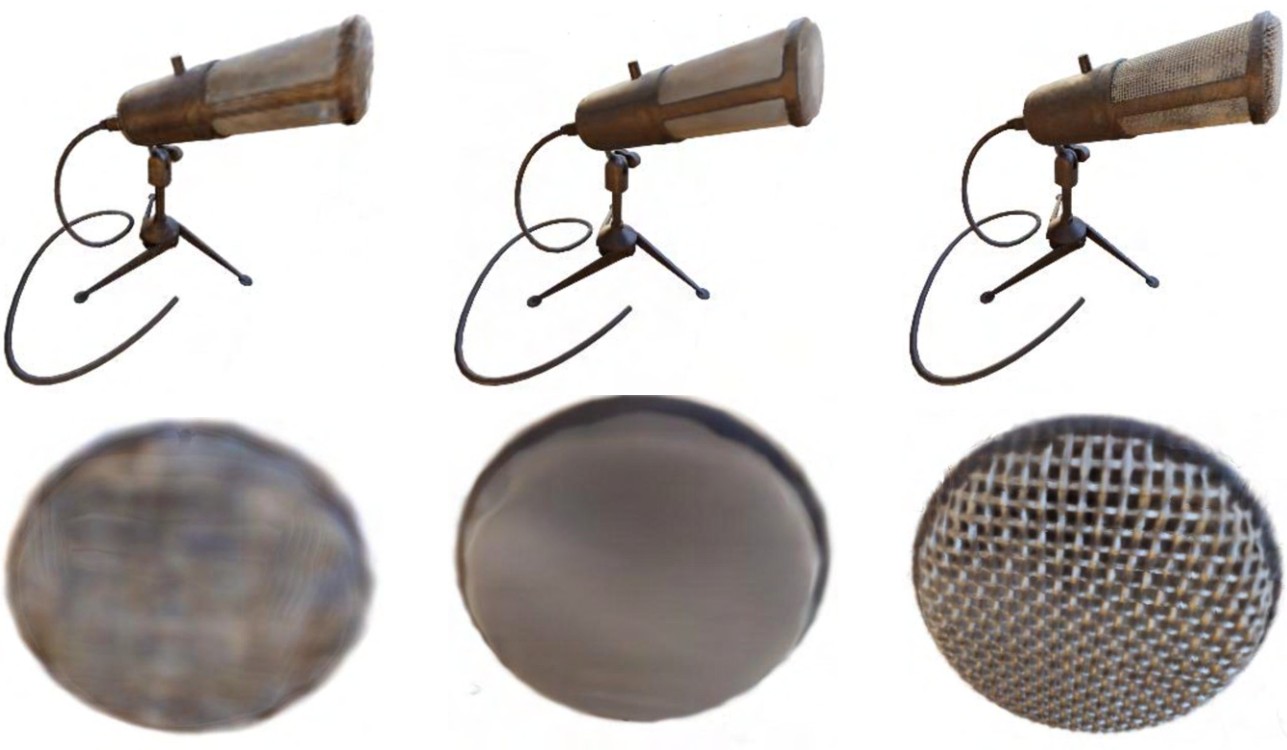

**Figure 17:** Rendering comparison for the *mic* scene: TensoRF on the left (0.32 M parameters), K-Planes in the middle (0.39 M parameters), GA-Planes on the right (0.25 M parameters).

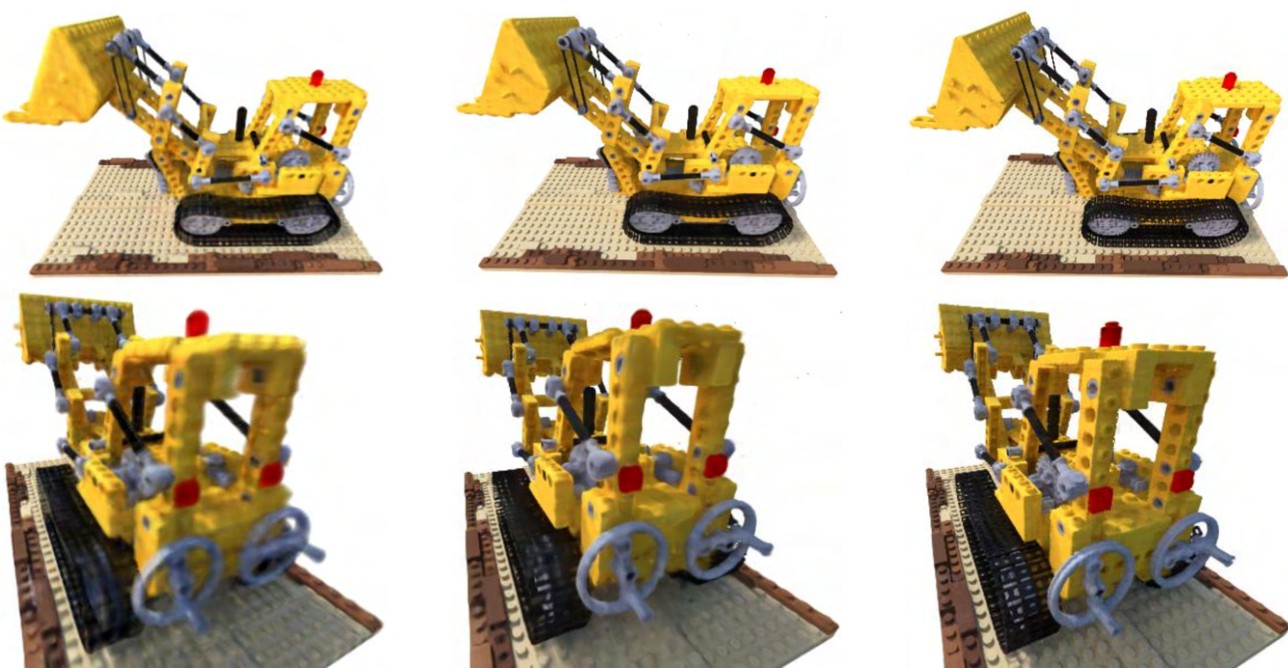

**Figure 18:** Rendering comparison for the *lego* scene: TensoRF on the left (0.32 M parameters), K-Planes in the middle (0.39 M parameters), GA-Planes on the right (0.25 M parameters).

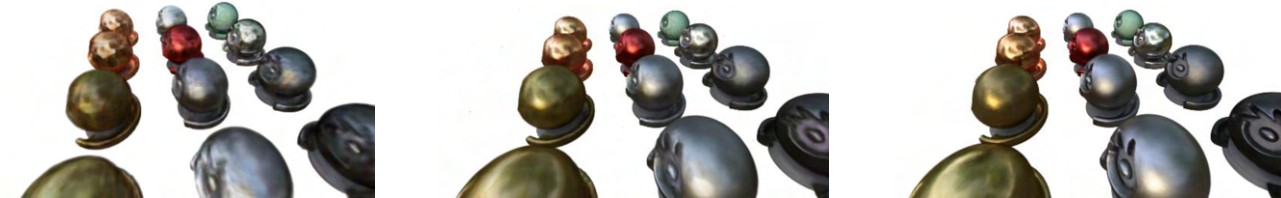

**Figure 19:** Rendering comparison for the *materials* scene: TensoRF on the left (0.32 M parameters), K-Planes in the middle (0.39 M parameters), GA-Planes on the right (0.25 M parameters).

| Model | Resolutions | Channel Dimensions | Multiresolution | Proposal Network Resolutions | Number of model parameters (M) |
|---|---|---|---|---|---|
| **K-plane** | 32 | 4 | [1, 2, 4] | [32, 64] | 0.390 |
| | 32 | 8 | [1, 2, 4] | [32, 64] | 0.649 |
| | 64 | 4 | [1, 2, 4] | [64, 128] | 1.533 |
| | 64 | 8 | [1, 2, 4] | [128, 256] | 4.041 |
| | 64 | 16 | [1, 2, 4] | [128, 256] | 6.107 |
| | 128 | 8 | [1, 2, 4] | [128, 256] | 10.234 |
| | 128 | 16 | [1, 2, 4] | [128, 256] | 18.493 |
| **K-plane without proposal sampling** | 32 | 4 | [1, 2, 4] | - | 0.298 |
| | 40 | 4 | [1, 2, 4] | - | 0.444 |
| | 64 | 4 | [1, 2, 4] | - | 1.073 |
| | 64 | 8 | [1, 2, 4] | - | 2.108 |
| | 100 | 6 | [1, 2, 4] | - | 3.822 |
| | 128 | 10 | [1, 2, 4] | - | 10.367 |
| | 129 | 16 | [1, 2, 4] | - | 16.824 |
| **TensoRF** | 128 | [2, 4] | - | - | 0.320 |
| | 256 | [2, 4] | - | - | 1.207 |
| | 256 | [6, 8] | - | - | 2.786 |
| | 256 | [12, 12] | - | - | 4.760 |
| | 300 | [12, 16] | - | - | 7.609 |
| | 300 | [16, 24] | - | - | 10.860 |
| | 300 | [32, 32] | - | - | 17.362 |
| | 300 | [16, 48] | - | - | 17.364 |
| **GA-plane** | [200, 4, 4] | [32, 32, 4] | [1, 2, 4] | - | 0.254 |
| | [200, 8, 4] | [32, 32, 4] | [1, 2, 4] | - | 0.351 |
| | [200, 16, 8] | [32, 32, 4] | [1, 2, 4] | - | 0.740 |
| | [200, 32, 8] | [16, 16, 4] | [1, 2, 4] | - | 1.164 |
| | [100, 100, 16] | [6, 6, 8] | [1, 2, 4] | - | 3.874 |
| | [200, 128, 32] | [10, 10, 8] | [1, 2, 4] | - | 10.681 |
| | [200, 128, 32] | [16, 16, 8] | [1, 2, 4] | - | 16.908 |
| **GA-plane ablation-VM** | 32 | 4 | [1, 2, 4] | - | 0.301 |
| | 64 | 4 | [1, 2, 4] | - | 1.078 |
| | 128 | 4 | [1, 2, 4] | - | 4.180 |
| | 128 | 6 | [1, 2, 4] | - | 6.251 |
| | 128 | 12 | [1, 2, 4] | - | 12.465 |
| **GA-plane ablation-CP** | 200 | 32 | [1, 2, 4] | - | 0.196 |
| | 200 | 64 | [1, 2, 4] | - | 0.355 |
| **GA-plane ablation-volume** | 18 | [3, 5, 6] | [1, 2, 3] | - | 1.236 |
| | 24 | [4, 4, 6] | [1, 2, 3] | - | 2.778 |
| | 32 | [4, 4, 6] | [1, 2, 3] | - | 6.529 |
| | 32 | [4, 6, 8] | [1, 2, 3] | - | 8.824 |

**Table 5:** Model configurations used for the radiance field modeling task on the Blender dataset.

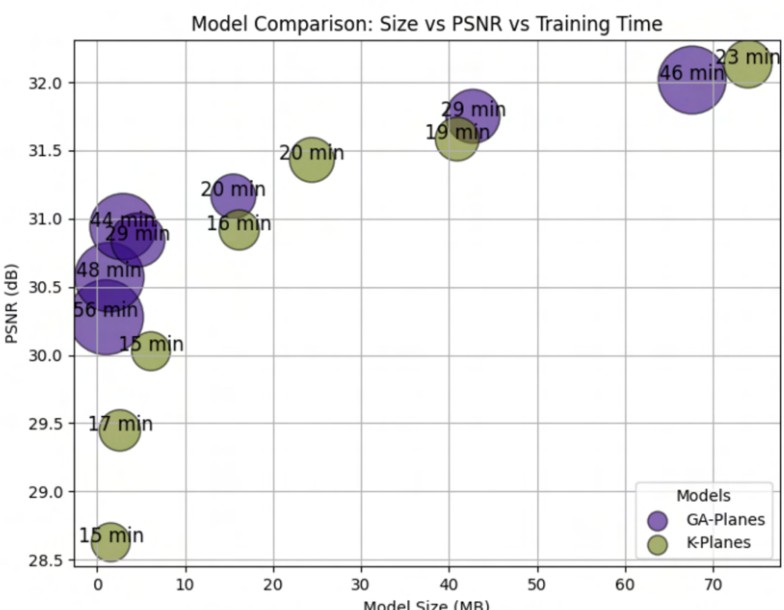

**Figure 20:** We plot the training time, memory and PSNR values of GA-Planes and the strongest baseline, K-Planes. The sizes of the points are proportional to training times, as written on the plot. Note that the training durations are directly affected by the selected feature dimensions, and GA-Planes can be configured accordingly to enable faster training (and indeed achieves the same speed as K-Planes if the feature dimensions are set to be equal). Here we show the times required for the model configurations used in radiance field reconstruction task.

