# OpenReview forum: "Geometric Algebra Planes: Convex Implicit Neural Volumes"
_ICML.cc/2025/Conference — ICML 2025 poster_

### Official Review · Reviewer_YmR7 · 2025-03-10

**Overall Recommendation:** 4

**Summary:**

The paper aims to improve grid-based representations for neural fields.
Inspired by principles from geometric algebra, the paper introduces a set of formulations, specifically, convex and semi-convex representations, to enhance the expressiveness and efficiency of grid-based neural fields.
Experimental results on several tasks, including 3D novel view synthesis, segmentation, and video segmentation, demonstrate that the proposed method can improve the efficiency of the representations.

## update after rebuttal
No major concerns were raised, and most of the concerns were resolved during rebuttal. Thus, I would like to keep my initial score.

**Claims And Evidence:**

The paper claims that incorporating geometric algebra concepts into grid-based neural field representations can enhance expressibility.
To support this, the authors provide experimental results on multiple tasks.

**Essential References Not Discussed:**

The paper cites all essential references.

**Experimental Designs Or Analyses:**

The experimental designs appear to be standard.

**Methods And Evaluation Criteria:**

Although the idea of combining lines, planes, and cubes for 3D representations is not new in this domain, the paper introduces intriguing idea of connecting them to Geometry algebra and convexity.
However, the transition to the introduction of convexity seems abrupt.
In addition, it is not entirely clear whether the choice among these formulations is task-dependent or if there is a general guideline for their application.

**Other Comments Or Suggestions:**

In Section 3.1, the paper states, "The specific models we use for nonconvex, semiconvex, and convex optimization are illustrated in Figure 1", yet only the nonconvex model is shown.

**Other Strengths And Weaknesses:**

No comments

**Questions For Authors:**

Q1: Is there a specific reason for including the overviews of the three architectures in the supplementary rather than in the main paper?
Q2: In Table 2, how is the performance of Tri-Planes model with concatenation?
Q3: In 2D representation, novel view synthesis, and novel view object segmentation, non-convex representations seem to perform best, while semi-convex seem to perform best for video segmentation tasks. Could you provide more detailed guidelines or discussion on how one should select among nonconvex, semi-convex, and convex representations for a given task?

**Relation To Broader Scientific Literature:**

This work is closely related to prior research on efficient grid-based representations, such as TensoRF, K-Planes, and MERF.
The proposed method could provide interesting direction to convexity and other untouched aspects in this domain.

**Theoretical Claims:**

Theoretical claims seem valid.
The analysis, based on the 2D setting, is extended to 3D tasks through experiments.

---

> ### Author Rebuttal · Authors · 2025-04-01
>
> We thank the reviewer for their thoughtful engagement with our work.
>
> Response to other comments and suggestions: Thank you for noticing the inconsistency with figure 1. This is a typo, the full figure is in the appendix (figure 6) due to space limitations in the main text.
>
> Response to questions:
>
> Q1 (full architecture figure in appendix): We did not have enough space.
>
> Q2 (performance of Tri-Planes with concatenation): The performance is very similar to original Tri-Planes (with feature addition). For instance, in the 3D segmentation with 2D supervision task (task 1), Tri-Planes with concatenation using the same channel dimensions and resolutions (leading to slightly more parameters as the input feature size to the MLP decoder is increased) achieves the following IOU results: 0.691 (convex), 0.870 (semiconvex), 0.868 (nonconvex). The results achieved by addition of the features were: 0.681 (convex), 0.868 (semiconvex), 0.863 (nonconvex). Since the difference is very minor, we present Tri-Planes in its original form.
>
> Q3 (guidelines for when to use convex, semi-convex, vs. nonconvex GA-Planes): The radiance field reconstruction task is inherently nonconvex due to the nonlinear forward model. Thus, using a semiconvex or convex model does not provide any theoretical or practical advantages. For the other 2 tasks, convexity and semi-convexity improve stability with respect to random initialization and training procedure. However, empirically we find that nonconvex models also work well once model size is large enough; convexity is most essential for small models that are most sensitive to random initialization. Convex models can also be trained very quickly with specialized convex solvers (a future direction for our project). Semiconvex models don’t have fast, dedicated solvers like convex models, but they also achieve the global minimum of the objective and are thus more stable compared to nonconvex models with similar performance.

---

### Official Review · Reviewer_BtXY · 2025-03-14

**Overall Recommendation:** 3

**Summary:**

The paper reviews existing literature on INRs, noticing that each method presents a trade off between its representation power and its size and optimizability. Based on Clifford algebra, Geometric Algebra Planes are introduced, generalizing some of the existing approaches which use 2 or 3-dimensional feature grids. GA uses a combination of grids at different dimensions and resolutions, and using different combination methods (e.g. multiplication and/or concatenation), resulting in convex, semi-convex and non-convex models. A theoretical derivation describes the behaviour and bounds of the proposed approach, also providing practically useful insights. Experiments with neural radiance fields and segmentation are shown to validate the method.

#Update after rebuttal
The rebuttal addressed most of my concerns. I am still not convinced of the validity of the video segmentation experiment, but overall I lean towards acceptance.

**Claims And Evidence:**

The claims are mostly supported by evidence. Experimental evidence could be improved (see below).

**Essential References Not Discussed:**

Potential additional works needed for the segmentation experiments (see above).
A related works section on Clifford algebra-related methods should also be included [2] [3] [4], given the claim of being "the first use of GA in neural volume models"

[2] Clifford Neural Layers for PDE Modeling, Johannes Brandstetter, Rianne van den Berg, Max Welling, Jayesh K. Gupta
[3] Clifford Group Equivariant Neural Networks, Ruhe et al.
[4] Geometric Clifford Algebra Networks, Ruhe et al.

**Experimental Designs Or Analyses:**

The experiment on NeRF is valid, and shows a comparison with the most important baselines, showing good performance across multiple model sizes.
E1) The 3D segmentation experiment also seem valid and show that GA-planes supersedes Tri-planes. However, to fully convince the reader, other baselines should be included as well. Examples can be found in [1]. While achieving state of the art might not be the goal of the work, showing that the GA-planes can improve existing works seems to be beneficial. Can GA-planes be used in conjunction with existing models used as (probably non-convex) decoders?
E2) The video segmentation task is unclear: it shows that GA-planes is better than tri-planes, however I do not understand the significance of a segmentation task where the ground truth comes from an existing segmentation model.

E3) Experiments on shape/image representation could also show the benefit of the method, and would have been an important addition to the paper to show the general validity of the method.


[1] Deep Learning Based 3D Segmentation: A Survey, Yong He, Hongshan Yu, Xiaoyan Liu, Zhengeng Yang, Wei Sun, Saeed Anwar, Ajmal Mian

**Methods And Evaluation Criteria:**

The proposed method appears appropriate for the described problem.

**Other Comments Or Suggestions:**

Figure 1 is referenced in the text as containing 3 rows but it only has one. The text is referring to the supplementary, I imagine. Having a full (but smaller) figure in the main text would be beneficial.

**Other Strengths And Weaknesses:**

Strengths:
S1) The paper is well written
S2) The analyses are well made and offer good insights with also practical applications, as in section 4.2. Figure 3 is also very informative, including multiple methods at different sizes.
S3) The method seems well grounded and generalises common approaches such as Tri-planes

Weaknesses:
W1) The experimental weaknesses raised above concern me about the practical usability of the method, so I would like the authors to address those concerns.
W2) Does the method come with a speed and/or memory overhead? Given the claims of performance/memory optimality, this should be addressed.

**Questions For Authors:**

I would like the authors to address my concerns about experimental design and memory/speed overhead.

**Relation To Broader Scientific Literature:**

The contributions seem sufficiently contextualised, both in their theoretical framework and among recent baselines, except on segmentation (see concerns above).

**Theoretical Claims:**

I am familiar with the described problem, but not an expert in the theory behind it, therefore I am unable to verify the proofs for the theoretical claims.

---

> ### Author Rebuttal · Authors · 2025-04-01
>
> Thank you for your comments and thoughtful engagement with our work. We address your concerns below.
>
> Response to E1 (other baselines and nonconvex decoders): Thank you for referencing [1]. In our 3D segmentation experiments, we use a different dataset compared to data types mentioned by the survey paper. The task we consider is to lift the 2D segmented images of a particular scene to get the 3D segmentation of the object, which is very similar to the normal radiance field reconstruction task, with the only difference being the lack of color prediction. We chose this task as it has a linear forward operation and can be formulated as a convex optimization problem with the use of a convex loss function (like mean-squared error). In the referenced paper, the segmentation is done on different representations of the 3D space. By focusing on optimizing a method for reconstructing 3D scenes from 2D images, we allow for the use of any highly performant, open source pre-trained segmentation model to be used to produce binary masks to be lifted to 3D by our method. Thus, any improvement of 2D segmentation models can be directly reflected in our 3D segmentation output. Regarding decoders, our GA-Planes representation can be used as a feature embedding with any desired decoder including existing INRs as decoders; indeed our nonconvex GA-Planes model uses a standard nonconvex MLP decoder. For generative modeling, a similar approach like EG3D paper (using Tri-Plane representation) could be used with GA-Planes as the underlying representation (this is an exciting direction for future work).
>
> Response to E2 (clarification on video segmentation): The video segmentation task we consider is the same as 3D segmentation with 2D+time (xyt) replacing 3D (xyz). We show performance on temporal superresolution in this task, which is interpolating missing frames, similar to rendering missing view angles. Again, here we are taking per-frame segmentation masks as training input and our goal is to essentially interpolate these 2D masks into a coherent 3D mask across the video (including missing test frames).
>
> Response to E3 (shape/image representation): Our 3D segmentation experiment is performing shape fitting (with either direct 3D supervision via space carving, or indirect 2D “tomographic” supervision); we will clarify our description of this task in the revision. We also include image fitting experiments to support our theoretical analysis (figure 7 in appendix of original submission).
>
> Response to GA references: Thank you for pointing us to these models using geometric algebra. We are happy to include them in our revised related works discussion.
>
> Response to weaknesses:
>
> W1 (clarification on existing experiments): Please refer to our responses above to E1, E2, and E3.
>
> W2 (speed and memory): No, our method does not come with a speed or memory overhead. We compare the number of parameters (memory) and performance metrics such as PSNR in figure 3. We also added a plot comparing memory and training times of GA-Planes and K-Planes with the configurations as in figure 3 (https://imgur.com/a/Kgm12Pk). Note that the variations in training time are directly influenced by the input feature dimension that gets decoded by the MLP. Comparing GA-Planes and K-Planes with the same feature dimensions, their training times are on par. As noticeable on the plot, GA-Planes had larger feature dimensions on some models, leading to longer training times (we selected the configurations to optimize memory vs. PSNR, which we list on appendix table 5).
>
> Response to suggestions: Thank you for noticing the typo about referring to figure 1. We will put the full figure in the main text.

---

> > ### Comment · Reviewer_BtXY · 2025-04-08
> >
> > Thank you for your response. My doubts are mostly cleared, except for the video segmentation.
> > To my understanding, an existing segmentation model (SAM) is used as GT. Why is it significant to show that GA planes can mimic an existing model better than Tri-Planes?

---

> > > ### Author Response · Authors · 2025-04-09
> > >
> > > Thank you for your response.
> > >
> > > To clarify, the task we are testing is temporal superresolution of segmentation masks. SAM is producing a segmentation mask for each frame in the video, which can be treated as the data preparation stage. We are trying to render the segmented video from a subset of the segmentation masks, which involves interpolating masks for the frames in between given frames. It’s analogous to our 3D object segmentation task where we are also not trying to create a segmentation mask from raw images but instead trying to lift 2D segmentation masks to 3D segmentations. Interpolation of unseen frames (temporal superresolution) is analogous to rendering images for unseen view angles.
> > >
> > > The point of comparing GA-Planes and Tri-Planes is to demonstrate the necessity of the GA-Planes parameterization to enable convex and semiconvex training with good performance. Tri-Planes is the most similar previously proposed model to ours that is compatible with convex reformulation, and we show that the convex formulation for GA-Planes performs consistently better in both video and 3D segmentation tasks.

---

### Official Review · Reviewer_hYwc · 2025-03-15

**Overall Recommendation:** 4

**Summary:**

This paper provides an analysis of the mixture of the n-dimensional (n<3) voxel representations for learning neural fields. As the authors mentioned, this voxel representation can be line, plane, or volume, which can be viewed as a low-rank or low-resolution representation to encapsulate the target scenes or images. By providing 4 theorems, the authors analyze the expressiveness of each different model design which is consistent with the experimental results. Moreover, in my understanding, this analysis and the results are also consistent with the related studies, such as K-Planes, TensorF, etc.

**Claims And Evidence:**

Yes. The claims in the paper are precise and correct. The authors provide four theorems that progressively investigate the expressiveness of the models in terms of the matrix rank. The first theorem is the most basic model that simply consists of line grids with a linear decoder, which is close to the matrix decomposition as the authors mentioned. The last theorem is the combination of the low-rank and low-resolution grids with non-linear MLP layers that are understood as the most expressive representations compared to the previous models explained in previous theorems. I fully understand these theorems and I think it is correct and well analyzed.

**Essential References Not Discussed:**

Overall, this paper well addresses the generic issue in the hybrid grid representation. I cannot find anything that the authors missed. In the supplementary material, the authors provide additional results and technical comparison with various methods, such as the hash-grid-based method [InstantNGP]. So, I believe there is nothing to discuss more.

**Ethical Review Concerns:**

There is no ethical issue in this paper.

**Experimental Designs Or Analyses:**

The authors conduct (1) image reconstruction in Figure 2, (2) radiance field modeling in Figure 3, and 2D/3D/Video segmentation in Table 2. By doing so, the authors well prove their claim about the model expressiveness in different domains, which looks sound and reasonable to me. Overall, the authors provide qualitative and quantitative results, and the results well support the author's claim.

Especially. the results in Figure3 well explain the necessity of the low-rank and low-resolution grid representation. The authors compare the proposed GA model with two popular low-rank grid-based methods, TensorF, and K-planes, and the results from the proposed method outperform the previous methods. This is a strong clue to support the authors' claims.

**Methods And Evaluation Criteria:**

Based on these four theorems, the authors provide a GA model that represents n-dimensional data (2D images or 3D scenes) with low-rank and low-resolution grid representation with non-convex MLP decoders. Figure 2 and Figure 3 support the efficacy of this method and the evaluation metrics are properly chosen.

Moreover, the authors conduct 2D/3D segmentation experiments in Section 5.2, and video segmentation in Section 5.3. By doing so, the authors provide supportive results in 2D/3D/2D+temporal domains. While there are no experiments handling n-dimensional data, it is fine with me.

**Other Comments Or Suggestions:**

I described my questions in the above section. Please refer to the one.

**Other Strengths And Weaknesses:**

This paper is well written. Providing four theorems is highly admirable and I really enjoyed reading this manuscript. It well supports the results in this paper as well as the results in many recent studies, which claim why hybrid grid representation can be beneficial to their downstream tasks.

I have a few concerns about this work.

(1) Can the authors provide a comparison with volume volume-based method as well in Figure 3? Theoretically speaking, the full-volume grid representation can be the most expressive compared to other grid-based methods including the proposed GA model. However, it is widely known that the volume-only methods tend to fall into the local minima with relatively low fidelity results compared to the MLP-only methods, or MLP-voxel hybrid methods [MipNeRF, ZipNeRF]. For example in the paper of [MonoSDF]. The MLP-only methods outperform the volume grid-based methods in the surface reconstruction task.

(2) In this perspective, I believe the low-rank problem can become a rescue to avoid falling into such a problem. While expressiveness is an important factor, it is not the only one that affects the final rendering quality in my opinion. __I hope that the authors provide an opinion about this issue in the rebuttal.__

(3) I wonder whether we can provide the expressiveness of the MLP-only method, such as Mip-NeRF. Can the authors provide their own analysis of this method?

**Questions For Authors:**

I described my questions in the above section. Please refer to the one.

**Relation To Broader Scientific Literature:**

So far, the recent studies in the neural fields using low-rank and low-resolution grids only provide empirical results without theoretical analysis in their model designs. At this point, this paper provide profound and theoretical reasons that align with the recent studies. I believe that this paper well describe their understanding in the hybrid grid representation and this will be beneficial to the researchers in this domain.

**Theoretical Claims:**

Overall, the theorems are correct. In the first theorem, the authors present the basic model using two matrices U, V which is the same as matrix decomposition. The second theorem is the replacement of matrix multiplication with matrix addition. This is to provide an analysis that the matrix addition has the limitation of expressiveness due to the upper bounded matrix rank. The third theorem is an extension of the first theorem using two-layered MLP decoder. The last theorem is the generalized version of the third theorem, which utilizes the standard MLP layers.

For each step, the authors provide theoretical analysis in terms of the model expressiveness by checking the matrix rank, and these results are summarized in Table 1 of the manuscript.

---

> ### Author Rebuttal · Authors · 2025-04-01
>
> Thank you for your thorough review.
>
> Response to 1 (comparison with volume-based method): The pink dashed line marked as “GA-Planes ablation (volume only)” is a volume-based method. In the notation we use, this is D($e_{123}$). We believe the inferior performance is caused by the coarser resolution of the 3D grid compared to the higher resolution that 2D and 1D grids can afford with the same total parameter count. Increasing the resolution of the volume features is expensive in terms of memory. Based on our 2D theory, the use of an MLP decoder enables high effective rank (limited by the resolution) representations even through “low-rank” components such as lines. Thus, the volume-only model does not achieve the expressivity of a combination of high-resolution lines/planes at the same memory budget.
>
> Response to 2 (local minima vs model expressiveness for volume-based vs factorized and implicit models): We note that for NeRF training, the objective function itself is nonconvex, so regardless of the volume parameterization there is a risk of getting stuck in a local minimum. However, if we consider convex objectives such as shape/video/image fitting, then a 3D grid or a concatenation of tensor features (e.g. our convex GA-Planes model) should be able to optimize globally, whereas a tensor factorization involving feature products (e.g. our nonconvex GA-Planes model, and K-Planes) would still be nonconvex. As described above, we believe that the primary reason behind the performance improvement we observe compared to a 3D grid is based on the difference in resolution, which improves the representation capacity or effective rank of our models, rather than a difference in optimizability.
>
> Response to 3 (analysis of MLP-based models): With a purely MLP-based model, we cannot define an equivalent matrix factorization optimization problem like the ones we provide in our theoretical analysis of GA-Planes. However, some prior works [1] have analyzed the NeRF model with Fourier features through the lens of neural tangent kernels (NTKs).
> [1] https://arxiv.org/pdf/2006.10739

---

> > ### Comment · Reviewer_hYwc · 2025-04-08
> >
> > Thank you for the rebuttal. Most of my concerns are clearly answered and I fully understand the authors' opinions. I have no further questions and I keep my original score, accept, as a final score. Thanks for the sincere and precise comments.
> >
> > Best,

---

### Decision · Program_Chairs · 2025-05-01

**Decision:**

Accept (poster)

**Comment:**

The submission got 3 positive recommendations eventually. The reviewers were initially concerned about the evaluations, technical details, and the analysis. The authors did a good job in their rebuttal and addressed most of these concerns. The reviewers did not actively engage in a discussion although the AC tried to initiate one. The AC read through the submission, all reviews, and rebuttals. The AC agreed with the reviewers and values the idea. Per this, the AC made a decision of acceptance. The decision was also approved by the senior AC.